# Accelerating ERM for data-driven algorithm design using output-sensitive techniques

**Maria-Florina Balcan**[*]      **Christopher Seiler**[†]      **Dravyansh Sharma**[‡]

## Abstract

Data-driven algorithm design is a promising, learning-based approach for beyond worst-case analysis of algorithms with tunable parameters. An important open problem is the design of computationally efficient data-driven algorithms for combinatorial algorithm families with multiple parameters. As one fixes the problem instance and varies the parameters, the "dual" loss function typically has a piecewise-decomposable structure, i.e. is well-behaved except at certain sharp transition boundaries. Motivated by prior empirical work, we initiate the study of techniques to develop efficient ERM learning algorithms for data-driven algorithm design by enumerating the pieces of the sum dual loss functions for a collection of problem instances. The running time of our approach scales with the actual number of pieces that appear as opposed to worst case upper bounds on the number of pieces. Our approach involves two novel ingredients – an output-sensitive algorithm for enumerating polytopes induced by a set of hyperplanes using tools from computational geometry, and an *execution graph* which compactly represents all the states the algorithm could attain for all possible parameter values. We illustrate our techniques by giving algorithms for pricing problems, linkage-based clustering and dynamic-programming based sequence alignment.

## 1 Introduction

The data-driven algorithm design paradigm captures a widely occuring scenario of solving multiple related problem instances and allows the design and analysis of algorithms that use machine learning to learn how to solve the instances which come from the same domain [GR16, Bal20]. Typically there are large (often infinite) parameterized algorithm families to choose from, and data-driven algorithm design approaches provide techniques to select algorithm parameters that provably perform well for instances from the same domain. Data-driven algorithms have been proposed and analyzed for a variety of combinatorial problems, including clustering, computational biology and mechanism design [BDL20, BDD+21, BPS20]. But most of the prior work has focused on *sample efficiency* of learning good algorithms i.e. the number of problem instances needed to learn algorithm parameters that perform well on a typical problem from the domain. A major open question for this line of research is to design *computationally efficient* learning algorithms [GR20, BDS21].

The parameterized family may occur naturally in well-known algorithms used in practice, or one could potentially design new families interpolating known heuristics. For the problem of aligning pairs of genomic sequences, one typically obtains the best alignment using a dynamic program with some costs or weights assigned to edits of different kinds, such as insertions, substitutions, or reduplications [Wat89]. These costs are the natural parameters for the alignment algorithm. The quality of the alignment can strongly depend on the parameters, and the best parameters vary depending on the application (e.g. aligning DNAs or RNAs), the pair of species being compared, or

---

[*]Carnegie Mellon University, `ninamf@cs.cmu.edu`.

[†]Work done by Christopher Seiler while he was at CMU.

[‡]Corresponding author: `dravy@ttic.edu`. Work done by Dravyansh Sharma while he was at CMU.

38th Conference on Neural Information Processing Systems (NeurIPS 2024).

| Problem | Dim. | $T_S$ (prior work) | $T_S$ (ours) | $T_{\text{ERM}}$ ($m$ instances) |
|---|---|---|---|---|
| Linkage-based clustering | $d=2$ | $O(n^{18}\log n)$ [BDL20] | $O(Rn^3)$ | $mT_S + \tilde{O}(mn^2 R_\Sigma)$ |
| | any $d$ | $O(n^{8d+2}\log n)$ [BDL20] | $\tilde{O}(R^2 n^3)$ | $mT_S + \tilde{O}(mn^2 R_\Sigma^2)$ |
| DP-based sequence alignment | $d=2$ | $O(R^2 + RT_{\text{DP}})$ [GBN94] | $O(RT_{\text{DP}})$ | $mT_S + \tilde{O}(mT_{\text{DP}}R_\Sigma)$ |
| | any $d$ | $s^{O(sd)}T_{\text{DP}}$ [BDD$^+$21] | $\tilde{O}(\tilde{R}^{2L+1}T_{\text{DP}})$ | $mT_S + \tilde{O}(mT_{\text{DP}}R_\Sigma^2)$ |
| Two-part tariff pricing | $\ell=1$ | $O(K^3)$ [BPS20] | $\tilde{O}(R+K)$ | $O(R_\Sigma + mK\log mK)$ |
| | any $L'$ | $K^{O(L')}$ [BPS20] | $\tilde{O}(R^2 K)$ | $mT_S + \tilde{O}(mKR_\Sigma^2)$ |

Table 1: Summary of running times of the proposed algorithms. $T_{\text{ERM}}$ denotes the running time for computing the pieces in the sum dual class function in terms of $T_S$, the time for enumerating the pieces on a single problem instance (Theorem 2.3). $R_\Sigma$ (resp. $R$) denotes the number of pieces in the sum dual class function for given $m$ problem instances (resp. dual class function for a single problem instance). $n$ is the size of the clustering instance. $s$ is the length of sequences to be aligned, $L$ is the maximum number of subproblems in the sequence alignment DP update, $\tilde{R}$ is the maximum number of pieces in the dual class function over all subproblems, $T_{DP}$ is the time to solve the DP on a single instance. $K$ is the number of units of the item sold and $\ell$ is the menu length. The $\tilde{O}$ notation suppresses logarithmic terms and multiplicative terms that only depend on $d$ or $L$.

the purpose of the alignment. Similarly, item prices are natural parameters in automated mechanism design [BSV18, BPS20]. On the other hand, for linkage-based clustering, one usually chooses from a set of different available heuristics, such as single or complete linkage. Using an interpolation of these heuristics to design a parameterized family and tune the parameter, one can often obtain significant improvements in the quality of clustering [BNVW17, BDL20].

A common property satisfied by a large number of interesting parameterized algorithm families is that the loss[4] as a function of the real-valued parameters for any fixed problem instance—called the "dual class function" [BS21]—is a piecewise structured function, i.e. the parameter space can be partitioned into "pieces" via sharp transition boundaries such that the loss function is well-behaved (e.g. constant or linear) within each piece [Bal20]. Prior work on data-driven algorithm design has largely focused on the sample complexity of the empirical risk minimization (ERM) algorithm which finds the loss-minimizing value of the parameter over a collection of problem instances drawn from some fixed unknown distribution. The ERM on a collection of problem instances can be implemented by enumerating the pieces of the sum dual class loss function. We will design algorithms for computationally efficient enumeration of these pieces, when the transition boundaries are linear, and our techniques can be used to learn good domain-specific values of these parameters given access to multiple problem instances from the problem domain. More precisely, we use techniques from computational geometry to obtain "output-sensitive" algorithms (formalized in Appendix C), that scale with the output-size $R_\Sigma$ (i.e. number of pieces in the partition) of the piece enumeration problem for the sum dual class function on a collection of problem instances. Often, on commonly occurring problem instances, the number of pieces is much smaller than worst case bounds on it [BDL20, GS96] and our results imply significant gains in running time whenever $R_\Sigma$ is small.

**Our contributions.** We design novel approaches that use tools from computational geometry and lead to output-sensitive algorithms for learning good parameters by implementing the ERM (Empirical Risk Minimization) for several distinct data-driven design problems. The resulting learning algorithms scale polynomially with the number of sum dual class function pieces $R_\Sigma$ in the worst case (See Table 1) and are efficient for small constant $d$.

1. We present a novel output-sensitive algorithm for enumerating the cells induced by a collection of hyperplanes. Our approach applies to any problem where the loss is piecewise linear, with convex polytopic piece boundaries. We achieve output-polynomial time by removing redundant constraints in any polytope using Clarkson's algorithm (from computational geometry) [Cla94] and performing an implicit search over the graph of neighboring polytopes (formally Definition 2). Our results are useful in obtaining output-sensitive running times for several distinct data-driven algorithm design problems. Theorem 2.3 bounds the running time of the implementation of the

---

[4]or utitity, basically a function that measures the performance of any algorithm in the family on any problem instance.

ERM for piecewise-structured duals with linear boundaries (Definition 1) that scales with $R_\Sigma$ provided we can enumerate the pieces of the dual class function for a single problem instance. It is therefore sufficient to enumerate the pieces of dual function for a single instance, for which Theorem 2.2 is useful.

2. We show how to learn multidimensional parameterized families of hierarchical clustering algorithms[5]. Our approach for enumerating the pieces of the dual class function on a single instance extends the *execution tree* based approach introduced for the much simpler single-parameter family in [BDL20] to multiple dimensions. The key idea is to track the convex polytopic subdivisions of the parameter space corresponding to a single merge step in the linkage procedure via nodes of the execution tree and apply our output-sensitive cell-enumeration algorithm at each node.

3. For dynamic programming (DP) based sequence alignment, our approach for enumerating pieces of the dual class function extends the execution tree to an *execution directed acyclic graph* (DAG). For a small fixed $d$, our algorithm is efficient for small constant $L$, the maximum number of subproblems needed to compute any single DP update. Prior work [GBN94] only provides an algorithm for computing the full partition for $d = 2$, and the naive approach of comparing all pairs of alignments takes exponential time.

Our main contribution is to show that when the number of pieces ($R_\Sigma$ in Table 1) is small we can improve over computational efficiency of prior work. Our techniques involves a novel application of Clarkson's algorithm (used to remove redundant inequalities from system of linear equations) to derive output-sensitive enumeration, and execution graphs which capture problem-specific structure.

**Motivation from prior empirical work.** For the clustering problem, we give theoretical bounds on the output-size $R$ depending on the specific parametric family considered (e.g. Lemmas I.1, I.2). These bounds imply a strict improvement even for worst-case $R$, but can be much faster for typical $R$. Indeed, prior empirical work indicates $R$ is much smaller than the worst case bounds in practice [BDL20], where even for $d = 1$ there is a dramatic speed up of over $10^{15}$ times by being output-sensitive. For the sequence alignment problem, again the upper bounds on R depend on the nature of cost functions involved. For example, Theorem 2.2 of [GBN94] gives an upper bound on $R$ for the 2-parameter problem with substitutions and deletions. For the TPT pricing problem we prove new theoretical bounds on $R$ (Theorem G.3) that show worst-case improvements (cubic to quadratic) in the running time over prior work. In practice, $R$ is usually much smaller than the worst case bounds, making our results even stronger. In prior experimental work on computational biology data (sequence alignment with $d = 2$, sequence length ~100), [GS96] observe that of a possible more than $2^{200}$ alignments, only $R = 120$ appear as possible optimal sequence alignments (over $\mathbb{R}^2$) for a pair of immunoglobin sequences (a speed-up of over 58 orders of magnitude).

**Preliminaries.** We follow the notation of [Bal20]. For a given algorithmic problem (say clustering, or sequence alignment), let $\Pi$ denote the set of problem instances of interest. We also fix a (potentially infinite) family of algorithms $\mathcal{A}$, parameterized by a set $\mathcal{P} \subseteq \mathbb{R}^d$. Here $d$ is the number of real parameters, and will also be the 'fixed' parameter in the FPT (Fixed Parameter Tractability) sense (Appendix C). Let $A_\rho$ denote the algorithm in the family $\mathcal{A}$ parameterized by $\rho \in \mathcal{P}$. The performance of any algorithm on any problem instance is given by a utility (or loss) function $u : \Pi \times \mathcal{P} \to [0, H]$, i.e. $u(x, \rho)$ measures the performance on problem instance $x \in \Pi$ of algorithm $A_\rho \in \mathcal{A}$. The utility of a fixed algorithm $A_\rho$ from the family is given by $u_\rho : \Pi \to [0, H]$, with $u_\rho(x) = u(x, \rho)$. We are interested in the structure of the *dual class* of functions $u_x : \mathcal{P} \to [0, H]$, with $u_x(\rho) = u_\rho(x)$, which measure the performance of all algorithms of the family for a fixed problem instance $x \in \Pi$. We will use $\mathbb{I}\{\cdot\}$ to denote the 0-1 valued indicator function. For many parameterized algorithms, the dual class functions are piecewise structured in the following sense [BDD+21].

**Definition 1** (Piecewise structured with linear boundaries). *A function class $\mathcal{H} \subseteq \mathbb{R}^\mathcal{P}$ that maps a domain $\mathcal{P} \subseteq \mathbb{R}^d$ to $\mathbb{R}$ is $(\mathcal{F}, t)$-piecewise decomposable for a class $\mathcal{F} \subseteq \mathbb{R}^\mathcal{P}$ of piece functions if the following holds: for every $h \in \mathcal{H}$, there are $t$ linear threshold functions $g_1, \ldots, g_t : \mathcal{P} \to \{0, 1\}$, i.e. $g_i(x) = \mathbb{I}\{a_i^T x + b_i\}$ and a piece function $f_\mathbf{b} \in \mathcal{F}$ for each bit vector $\mathbf{b} \in \{0, 1\}^t$ such that for all $y \in \mathcal{P}$, $h(y) = f_{\mathbf{b}_y}(y)$ where $\mathbf{b}_y = (g_1(y), \ldots, g_t(y)) \in \{0, 1\}^t$.*

---

[5]One should carefully distinguish the parameterized clustering algorithm from the ERM-based learning algorithm which learns a parameter from this family given problem instances. Our algorithmic approaches focus on implementing the ERM efficiently.

We will refer to connected subsets of the parameter space where the dual class function is a fixed piece function $f_c$ as the *(dual) pieces* or *regions*, when the dual class function is clear from the context. Past work [BSV18] defines a similar definition for mechanism classes called $(d, t)$-*delineable* classes which are a special case of Definition 1, where the piece function class $\mathcal{F}$ consists of linear functions, and focus on sample complexity of learning, i.e. the number of problem instances from the problem distribution that are sufficient learn near-optimal parameters with high confidence over the draw of the sample. Our techniques apply for a larger class, where $\mathcal{F}$ is a class of convex functions. Past work [BSV18, BDL20, BDD$^+$21] provides *sample complexity* guarantees for problems that satisfy Definition 1, on the other hand we will focus on developing fast algorithms. We develop techniques that yield efficient algorithms for computing these pieces in the multiparameter setting, i.e. constant $d > 1$.The running times of our algorithms are polynomial in the output size (i.e, number of pieces). For $i \in \mathbb{Z}^+$, we will use $[i]$ to denote the set of positive integers $\{1, \dots, i\}$.

## 2   Output-sensitive cell enumeration and data-driven algorithm design

Suppose $H$ is a collection of $t$ hyperplanes in $\mathbb{R}^d$. We consider the problem of enumerating the $d$-faces (henceforth *cells*) of the convex polyhedral regions induced by the hyperplanes. The enumerated cells will be represented as sign-patterns[6] of facet-inducing hyperplanes. We will present an approach for enumerating these cells in OFPT time (Output-sensitive Fixed Parameter Tractable) Definition 3), which involves two key ingredients: (a) locality-sensitivity, and (b) output-sensitivity. By locality sensitivity, we mean that our algorithm exploits problem-specific local structure in the neighborhood of each cell to work with a smaller candidate set of hyperplanes which can potentially constitute the cell facets. This is abstracted out as a sub-routine COMPUTELOCALLYRELEVANTSEPARATORS which we will instantiate and analyse for each individual problem. In this section we will focus more on the output-sensitivity aspect.

To provide an output-sensitive guarantee for this enumeration problem, we compute only the *non-redundant* hyperplanes which provide the boundary of each cell $c$ in the partition induced by the hyperplanes. We denote the closed polytope bounding cell $c$ by $P_c$. A crucial ingredient for ensuring good output-sensitive runtime of our algorithm is Clarkson's algorithm for computing non-redundant constraints in a system of linear inequalities [Cla94]. A constraint is *redundant* in a system if removing it does not change the set of solutions. The key idea is to maintain a set $I$ of non-redundant constraints detected so far, and solve LPs that detect the redundancy of a remaining constraint (not in $I$) when added to $I$. If the constraint is redundant relative to $I$, it must also be redundant in the full system, otherwise we can add a (potentially different) non-redundant constraint to $I$ (see Appendix D for details). The following runtime guarantee is known for the algorithm.

**Theorem 2.1** (Clarkson's algorithm). *Given a list $L$ of $k$ half-space constraints in $d$ dimensions, Clarkson's algorithm outputs the set $I \subseteq L$ of non-redundant constraints in $L$ in time $O(k \cdot \mathrm{LP}(d, |I| + 1))$, where $\mathrm{LP}(v, c)$ is the time for solving an LP with $v$ variables and $c$ constraints.*

Algorithm 1 uses AUGMENTEDCLARKSON, which modifies Clarkson's algorithm with some additional bookkeeping (details in Appendix D) to facilitate a search for neighboring regions in our algorithm, while retaining the same asymptotic runtime complexity. Effectively, our algorithm can be seen as a breadth-first search over an implicit underlying graph (Definition 2), where the neighbors (and some auxiliary useful information) are computed dynamically by AUGMENTEDCLARKSON.

**Definition 2** (Cell adjacency graph). *Define the cell adjacency graph for a set $H$ of hyperplanes in $\mathbb{R}^d$, written $G_H = (V_H, E_H)$, as:*

- *There is a vertex $v \in V_H$ for each cell in the partition $\tilde{C}$ of $\mathbb{R}^d$ induced by the hyperplanes;*

- *For $v, v' \in V_H$, add the edge $\{v, v'\}$ to $E_H$ if the corresponding polytopes intersect, i.e. $P_v \cap P_{v'} \neq \emptyset$.*

*This generalizes to a subdivision (Definition 6) of a polytope in $\mathbb{R}^d$.*

This allows us to state the following guarantee about the runtime of Algorithm 1. In the notation of Table 1, we have $R = |V_H|$ and $|E_H| = O(R^2)$.

**Theorem 2.2.** *Let $H$ be a set of $t$ hyperplanes in $\mathbb{R}^d$. Suppose that $|E_H| = E$ and $|V_H| = V$ in the cell adjacency graph $G_H = (V_H, E_H)$ of $H$; then if the domain $\mathcal{P}$ is bounded by $|P| \leq t$ hyperplanes,*

---

[6]For simplicity we will denote these by vectors of the form $\{0, 1, -1\}^t$ where non-zero co-ordinates correspond to signs of facet-inducing hyperplanes. Hash tables would be a practical data structure for implementation.

---

**Algorithm 1:** OUTPUTSENSITIVEPARTITIONSEARCH

---

1: **Input**: Set $H = \{\mathbf{a_i} \cdot \mathbf{x} = b_i\}_{i \in [t]}$ of $t$ hyperplanes in $\mathbb{R}^d$, convex polytopic domain $\mathcal{P} \subseteq \mathbb{R}^d$ given by a set of hyperplanes $P$, procedure COMPUTELOCALLYRELEVANTSEPARATORS$(\mathbf{x}, H)$ with $\mathbf{x} \in \mathbb{R}^d$.

    **Output**: Partition cells $\tilde{C} = \{\tilde{c}^{(j)}\}$, $c^{(j)} \in \{0, 1, -1\}^t$ with $|\tilde{c}_i^{(j)}| = 1$ iff $\mathbf{a_i} \cdot \mathbf{x} = b_i$ is a bounding hyperplane for cell $j$, and $\mathsf{sign}(\tilde{c}_i^{(j)}) = \mathsf{sign}(\mathbf{a_i} \cdot \mathbf{x_j} - b_i)$ for interior point $\mathbf{x_j}$, $\mathsf{sign}(\cdot) \in \{\pm 1\}$.

2:   $\mathbf{x}_1 \leftarrow$ an arbitrary point in $\mathbb{R}^d$ (assumed general position w.r.t. $H$)

3:   Cell $c^{(1)}$ with $\mathsf{sign}(c_i^{(1)}) = \mathsf{sign}(\mathbf{a_i} \cdot \mathbf{x_1} - b_i)$

4:   $\mathtt{q} \leftarrow$ empty queue; $\mathtt{q.enqueue}([c^{(1)}, \mathbf{x}_1])$

5:   $C \leftarrow \{\}; \tilde{C} \leftarrow \{\}$

6: **while** $\mathtt{q.non\_empty()}$ **do**

7:     $[c, \mathbf{x}] \leftarrow \mathtt{q.dequeue()}$

8:     Continue to next iteration if $c \in C$

9:     $C \leftarrow C \cup \{c\}$

10:    $\tilde{H} \leftarrow$ COMPUTELOCALLYRELEVANTSEPARATORS$(\mathbf{x}, H)$ ;       `/* subset of` `hyperplanes in H that can be facets for cell containing x */`

11:    $H' \leftarrow \{(-\mathsf{sign}(c_i)\mathbf{a_i} \cdot \mathbf{x}, -\mathsf{sign}(c_i)b_i) \mid \mathbf{a_i} \cdot \mathbf{x} = b_i \in \tilde{H}\} \cup P$

12:    $(\tilde{c}, \mathtt{neighbors}) \leftarrow$ AUGMENTEDCLARKSON$(\mathbf{x}, H', c)$;      `/* Algorithm 2 */`

13:    $\tilde{C} \leftarrow \tilde{C} \cup \{\tilde{c}\}$

14:    $\mathtt{q.enqueue}([c', \mathbf{x}'])$ for each $[c', \mathbf{x}'] \in \mathtt{neighbors}$

15: **return** $\tilde{C}$

---

*Algorithm 1 computes the set $V_H$ in time $\tilde{O}(dE + VT_{CLRS} + t_{LRS} \cdot \sum_{c \in V_H} \mathrm{LP}(d, |I_c| + 1))$, where $\mathrm{LP}(\mathrm{r}, \mathrm{s})$ denotes the time to solve an LP in $r$ variables and $s$ constraints, $I_c$ denotes the number of facets for cell $c \in V_H$, $T_{CLRS}$ denotes the running time of* COMPUTELOCALLYRELEVANTSEPARA-TORS *and $t_{LRS}$ denotes an upper bound on the number of locally relevant hyperplanes in Line 10.*

We illustrate the significance of output-sensitivity and locality-sensitivity in our results with examples.

**Example 1.** *The worst-case size of $V_H$ is $O(t^d)$ and standard (output-insensitive) enumeration algorithms for computing $V_H$ (e.g. [EG86, Xu20]) take $O(t^d)$ time even when the output size may be much smaller. For example, if $H$ is a collection of parallel planes in $\mathbb{R}^3$, the running time of these approaches is $O(t^3)$. Even a naive implementation of* COMPUTELOCALLYRELEVANTSEPARATORS *which always outputs the complete set $H$ gives a better runtime of $O(t^2)$. By employing a straightforward algorithm which binary searches the closest hyperplanes to $\mathbf{x}$ (in a pre-processed $H$) as* COMPUTELOCALLYRELEVANTSEPARATORS *we have $t_{LRS} = O(1)$ and $T_{CLRS} = \tilde{O}(1)$, and Algorithm 1 attains a running time of $\tilde{O}(t)$. Analogously, if $H$ is a collection of $t$ hyperplanes in $\mathbb{R}^d$ with $1 \leq k < d$ distinct unit normal vectors, then output-sensitivity improves the runtime from $O(t^d)$ to $O(t^{k+1})$, and locality-sensitivity can be used to further improve it to $\tilde{O}(t^k)$.*

**ERM in the statistical learning setting.** We now use Algorithm 1 to compute the sample mimimum (aka ERM, Empirical Risk Minimization) for the $(\mathcal{F}, t)$ piecewise-structured dual losses with linear boundaries (Definition 1) over a problem sample $S \in \Pi^m$, provided piece functions in $\mathcal{F}$ can be efficiently optimized over a polytope (typically the piece functions are constant or linear functions in our examples). Formally, we define search-ERM for a given parameterized algorithm family $\mathcal{A}$ with parameter space $\mathcal{P}$ and the dual class utility function being $(\mathcal{F}, t)$-piecewise decomposable (Definition 1) as follows: given a set of $m$ problem instances $S \in \Pi^m$, compute the pieces, i.e. a partition of the parameter space into connected subsets such that the utility function is a fixed piece function in $\mathcal{F}$ over each subset for each of the $m$ problem instances. The following result gives a recipe for efficiently solving the search-ERM problem provided we can efficiently compute the dual function pieces in individual problem instances, and the the number of pieces in the sum dual class function over the sample $S$ is not too large. The key idea is to apply Algorithm 1 for each problem instance, and once again for the search-ERM problem. In the notation of Table 1, we have $R_\Sigma = |V_S|$,

the number of vertices in the cell adjacency graph corresponding to the polytopic pieces in the sum utility function $\sum_{i=1}^{m} u_i$. $|E_S| = O(R_\Sigma)$ when $d = 2$ and $O(R_\Sigma^2)$ for general $d$.

**Theorem 2.3.** *Let $\tilde{C}_i$ denote the cells partitioning the polytopic parameter space $\mathcal{P} \subset \mathbb{R}^d$ corresponding to pieces of the dual class utility function $u_i$ on a single problem instance $x_i \in \Pi$, from a collection $s = \{x_1, \ldots, x_m\}$ of $m$ problem instances. Let $(V_S, E_S)$ be the cell adjacency graph corresponding to the polytopic pieces in the sum utility function $\sum_{i=1}^{m} u_i$. Then there is an algorithm for computing $V_S$ given the cells $\tilde{C}_i$ in time $\tilde{O}((d+m)|E_S| + mt_{LRS} \cdot \sum_{c \in V_S} \mathrm{LP}(d, |I_c| + 1))$, where $I_c$ denotes the number of facets for cell $c \in V_S$, and $t_{LRS}$ is the number of locally relevant hyperplanes in a single instance.*

An important consequence of the above result is an efficient output-sensitive algorithm for data-driven algorithm design when the dual class utility function is $(\mathcal{F}, t)$-piecewise decomposable. In the following sections, we will instantiate the above results for various data-driven parameter selection problems where the dual class functions are piecewise-structured with linear boundaries (Definition 1). Prior work [BSV18, BDL20, BDD⁺21] has shown polynomial bounds on the sample complexity of learning near-optimal parameters via the ERM algorithm for these problems in the statistical learning setting, i.e. the problem instances are drawn from a fixed unknown distribution. In other words, ERM over polynomially large sample size $m$ is sufficient for learning good parameters. In particular, we will design and analyze running time for problem-specific algorithms for computing locally relevant hyperplanes. Given Theorem 2.3, it will be sufficient to give an algorithm for computing the pieces of the dual class function for a single problem instance.

**Remark 1.** *Our results in this section directly imply efficient algorithms for pricing problems in mechanism design which are known to be $(\mathcal{F}, t)$-decomposable where $\mathcal{F}$ is the class of linear functions (summarized in Table 1, see Appendix F for details).*

# 3 Linkage-based clustering

Clustering data into groups of similar points is a fundamental tool in data analysis and unsupervised machine learning. A variety of clustering algorithms have been introduced and studied but it is not clear which algorithms will work best on specific tasks. Also the quality of clustering is heavily dependent on the distance metric used to compare data points. Interpolating multiple metrics and clustering heuristics can result in significantly better clustering [BDL20].

**Problem setup.** Let $\mathcal{X}$ be the data domain. A clustering instance from the domain consists of a point set $S = \{x_1, \ldots, x_n\} \subseteq \mathcal{X}$ and an (unknown) target clustering $\mathcal{C} = (C_1, \ldots, C_k)$, where the sets $C_1, \ldots, C_k$ partition $S$ into $k$ clusters. Linkage-based clustering algorithms output a hierarchical clustering of the input data, represented by a cluster tree. We measure the agreement of a cluster tree $T$ with the target clustering $\mathcal{C}$ in terms of the Hamming distance between $\mathcal{C}$ and the closest pruning of $T$ that partitions it into $k$ clusters (i.e., $k$ disjoint subtrees that contain all the leaves of $T$). More formally, the loss $\ell(T, \mathcal{C}) = \min_{P_1, \ldots, P_k} \min_{\sigma \in S_n} \frac{1}{|S|} \sum_{i=1}^{k} |C_i \setminus P_{\sigma_i}|$, where the first minimum is over all prunings $P_1, \ldots, P_k$ of the cluster tree $T$ into $k$ subtrees, and the second minimum is over all permutations of the $k$ cluster indices.

A merge function $D$ defines the distance between a pair of clusters $C_i, C_j \subseteq \mathcal{X}$ in terms of the pairwise point distances given by a metric $d$. Cluster pairs with smallest values of the merge function are merged first. For example, single linkage uses the merge function $D_{\mathrm{sgl}}(C_i, C_j; d) = \min_{a \in C_i, b \in C_j} d(a, b)$ and complete linkage uses $D_{\mathrm{cmpl}}(C_i, C_j; d) = \max_{a \in C_i, b \in C_j} d(a, b)$. Instead of using extreme points to measure the distance between pairs of clusters, one may also use more central points, e.g. we define *median linkage* as $D_{\mathrm{med}}(C_i, C_j; d) = \texttt{median}(\{d(a, b) \mid a \in C_i, b \in C_j\})$, where $\texttt{median}(\cdot)$ is the usual statistical median of an ordered set $S \subset \mathbb{R}^7$. Appendix H provides synthetic clustering instances where one of single, complete or median linkage leads to significantly better clustering than the other two, illustrating the need to learn an interpolated procedure. Single,

---

[7]$\texttt{median}(S)$ is the smallest element of $S$ such that $S$ has at most half its elements less than $\texttt{median}(S)$ and at most half its elements more than $\texttt{median}(S)$. For comparison, the more well-known average linkage is $D_{\mathrm{avg}}(C_i, C_j; d) = \texttt{mean}(\{d(a, b) \mid a \in C_i, b \in C_j\})$. We may also use geometric medians of clusters. For example, we can define *mediod linkage* as $D_{\mathrm{geomed}}(C_i, C_j; d) = d(\arg\min_{a \in C_i} \sum_{a' \in C_i} d(a, a'), \arg\min_{b \in C_j} \sum_{b' \in C_j} d(b, b'))$.

median and complete linkage are *2-point-based* (Definition 4, Appendix I), i.e. the merge function $D(A, B; d)$ only depends on the distance $d(a, b)$ for two points $(a, b) \in A \times B$.

**Parameterized algorithm families.** Let $\Delta = \{D_1, \ldots, D_l\}$ denote a finite family of merge functions (measure distances between clusters) and $\delta = \{d_1, \ldots, d_m\}$ be a finite collection of distance metrics (measure distances between points). We define a parameterized family of linkage-based clustering algorithms that allows us to learn both the merge function and the distance metric. It is given by the interpolated merge function $D_\alpha^\Delta(A, B; \delta) = \sum_{D_i \in \Delta, d_j \in \delta} \alpha_{i,j} D_i(A, B; d_j)$, where $\alpha = \{\alpha_{i,j} \mid i \in [l], j \in [m], \alpha_{i,j} \geq 0\}$. In order to ensure linear boundary functions for the dual class function, our interpolated merge function $D_\alpha^\Delta(A, B; \delta)$ takes all pairs of distance metrics and linkage procedures. Due to invariance under constant multiplicative factors, we can set $\sum_{i,j} \alpha_{i,j} = 1$ and obtain a set of parameters which allows $\alpha$ to be parameterized by $d = lm - 1$ values[8]. Define the parameter space $\mathcal{P} = \blacktriangle^d = \left\{\rho \in (\mathbb{R}^{\geq 0})^d \mid \sum_i \rho_i \leq 1\right\}$; for any $\rho \in \mathcal{P}$ we get $\alpha^{(\rho)} \in \mathbb{R}^{d+1}$ as $\alpha_i^{(\rho)} = \rho_i$ for $i \in [d]$, $\alpha_{d+1}^{(\rho)} = 1 - \sum_{i \in [d]} \rho_i$. We focus on learning the optimal $\rho \in \mathcal{P}$ for a single instance $(S, \mathcal{Y})$. With slight abuse of notation we will sometimes use $D_\rho^\Delta(A, B; \delta)$ to denote the interpolated merge function $D_{\alpha^{(\rho)}}^\Delta(A, B; \delta)$. As a special case we have the family $D_{\alpha^{(\rho)}}^\Delta(A, B; d_0)$ that interpolates merge functions (from set $\Delta$) for different linkage procedures but the same distance metric $d_0$. Another interesting family only interpolates the distance metrics, i.e. use a distance metric $d_\rho(a, b) = \sum_{d_j \in \delta} \alpha_j^{(\rho)} d_j(a, b)$ and use a fixed linkage procedure. We denote this by $D_\rho^1(A, B; \delta)$.

We will extend the *execution tree* approach introduced by [BDL20] which computes the pieces (intervals) of *single-parameter* linkage-based clustering. A formal treatment of the execution tree, and how it is extended to the multi-parameter setting, is deferred to Appendix I.1. Informally, for a single parameter, the execution tree is defined as the partition tree where each node represents an interval where the first $t$ merges are identical, and edges correspond to the subset relationship between intervals obtained by refinement from a single merge. The execution, i.e. the sequence of merges, is unique along any path of this tree. The same properties, i.e. refinement of the partition with each merge and correspondence to the algorithm's execution, continue to hold in the multidimensional case, but with convex polytopes instead of intervals (see Definition 5). Computing the children of any node of the execution tree corresponds to computing the subdivision of a convex polytope into polytopic cells where the next merge step is fixed. The children of any node of the execution tree can be computed using Algorithm 1. We compute the cells by following the neighbors, keeping track of the cluster pairs merged for the computed cells to avoid recomputation. For any single cell, we find the bounding polytope along with cluster pairs corresponding to neighboring cells by computing the tight constraints in a system of linear inequalities. Theorem 2.2 gives the runtime complexity of the proposed algorithm for computing the children of any node of the execution tree. It only remains to specify COMPUTELOCALLYRELEVANTSEPARATORS. For a given $\mathbf{x} = \rho$ we find the next merge candidate in time $O(dn^2)$ by computing the merge function $D_\rho^\Delta(A, B; \delta)$ for all pairs of candidate (unmerged) clusters $A, B$. If $(A^*, B^*)$ minimizes the merge function, the locally relevant hyperplanes are given by $D_\rho^\Delta(A^*, B^*; \delta) \leq D_\rho^\Delta(A', B'; \delta)$ for $(A', B') \neq (A^*, B^*)$ i.e. $t_{\text{LRS}} \leq n^2$. Using Theorem 2.2, we give the following bound for the overall runtime of the algorithm (soft-O notation suppresses logarithmic terms and multiplicative constants in $d$, proof in Appendix I.2).

**Theorem 3.1.** *Let $S$ be a clustering instance with $|S| = n$, and let $R_i = |\mathcal{P}_i|$ and $R = R_n$. Let $H_t = \left|\{(\mathcal{Q}_1, \mathcal{Q}_2) \in \mathcal{P}_t^2 \mid \mathcal{Q}_1 \cap \mathcal{Q}_2 \neq \emptyset\}\right|$ denote the total number of adjacencies between any two pieces of $\mathcal{P}_i$ and $H = H_n$. Then, the leaves of the execution tree on $S$ can be computed in time $\tilde{O}\left(\sum_{i=1}^n (H_i + R_i T_M)(n - i + 1)^2\right)$, where $T_M$ is the time to compute the merge function.*

In the case of single, median, and complete linkage, we may assume $T_M = O(d)$ by carefully maintaining a hashtable containing distances between every pair of clusters. Each merge requires overhead at most $O(n_t^2)$, $n_t = n - t$ being the number of unmerged clusters at the node at depth $t$, which is absorbed by the cost of solving the LP corresponding to the cell of the merge. We have the following corollary which states that our algorithm is output-linear for $d = 2$.

**Corollary 3.2.** *For $d = 2$ the leaves of the execution tree of any clustering instance $S$ with $|S| = n$ can be computed in time $O(RT_M n^3)$.*

Above results yield bounds on $T_S$, the enumeration time for dual function of the pieces in a single problem instance. Theorem 2.3 further implies bounds on the runtime of ERM (Table 1).

---

[8]In contrast, the parametric family in [BDL20] has $l + m - 2$ parameters but it does not satisfy Definition 1.

# 4 Dynamic Programming based sequence alignment

Sequence alignment is a fundamental combinatorial problem with applications to computational biology. For example, to compare two DNA, RNA or amino acid sequences the standard approach is to align two sequences to detect similar regions and compute the optimal alignment [NW70, Wat89, CB00]. However, the optimal alignment depends on the relative costs or weights used for specific substitutions, insertions/deletions, or *gaps* (consecutive deletions) in the sequences. Given a set of weights, the optimal alignment computation is typically a simple dynamic program. Our goal is to learn the weights, such that the alignment produced by the dynamic program has application-specific desirable properties.

**Problem setup.** Given a pair of sequences $s_1, s_2$ over some alphabet $\Sigma$ of lengths $m = |s_1|$ and $n = |s_2|$, and a 'space' character $- \notin \Sigma$, a space-extension $t$ of a sequence $s$ over $\Sigma$ is a sequence over $\Sigma \cup \{-\}$ such that removing all occurrences of $-$ in $t$ gives $s$. A global alignment (or simply alignment) of $s_1, s_2$ is a pair of sequences $t_1, t_2$ such that $|t_1| = |t_2|$, $t_1, t_2$ are space-extensions of $s_1, s_2$ respectively, and for no $1 \le i \le |t_1|$ we have $t_1[i] = t_2[i] = -$. Let $s[i]$ denote the $i$-th character of a sequence $s$ and $s[:i]$ denote the first $i$ characters of sequence $s$. For $1 \le i \le |t_1|$, if $t_1[i] = t_2[i]$ we call it a *match*. If $t_1[i] \neq t_2[i]$, and one of $t_1[i]$ or $t_2[i]$ is the character $-$ we call it a *space*, else it is a *mismatch*. A sequence of $-$ characters (in $t_1$ or $t_2$) is called a *gap*. Matches, mismatches, gaps and spaces are commonly used *features* of an alignment, i.e. functions that map sequences and their alignments $(s_1, s_2, t_1, t_2)$ to $\mathbb{Z}_{\ge 0}$ (for example, the number of spaces). A common measure of *cost* of an alignment is some linear combination of features. For example if there are $d$ features given by $l_k(\cdot)$, $k \in [d]$, the cost may be given by $c(s_1, s_2, t_1, t_2, \rho) = \sum_{k=1}^{d} \rho_k l_k(s_1, s_2, t_1, t_2)$ where $\rho = (\rho_1, \ldots, \rho_d)$ are the parameters that govern the relative weight of the features [KK06]. Let $\tau(s, s', \rho) = \operatorname{argmin}_{t_1, t_2} c(s, s', t_1, t_2, \rho)$ and $C(s, s', \rho) = \min_{t_1, t_2} c(s, s', t_1, t_2, \rho)$ denote the optimal alignment and its cost respectively.

**A general DP update rule.** For a fixed $\rho$, suppose the sequence alignment problem can be solved, i.e. we can find the alignment with the smallest cost, using a dynamic program $A_\rho$ with linear parameter dependence (described below). Our main application will be to the family of dynamic programs $A_\rho$ which compute the optimal alignment $\tau(s_1, s_2, \rho)$ given any pair of sequences $(s_1, s_2) \in \Sigma^m \times \Sigma^n = \Pi$ for any $\rho \in \mathbb{R}^d$, but we will proceed to provide a more general abstraction. See Section J.1 for example DPs using well-known features in computational biology, expressed using the abstraction below. For any problem $(s_1, s_2) \in \Pi$, the dynamic program $A_\rho$ ($\rho \in \mathcal{P} \subseteq \mathbb{R}^d$, the set of parameters) solves a set $\pi(s_1, s_2) = \{P_i \mid i \in [k], P_k = (s_1, s_2)\}$ of $k$ subproblems (typically, $\pi(s_1, s_2) \subseteq \Pi_{s_1, s_2} = \{(s_1[:i'], s_2[:j']) \mid i' \in \{0, \ldots, m\}, j' \in \{0, \ldots, n\}\} \subseteq \Pi$) in some fixed order $P_1, \ldots, P_k = (s_1, s_2)$. Crucially, the subproblems sequence $P_1, \ldots, P_k$ do not depend on $\rho$[9]. In particular, a problem $P_j$ can be efficiently solved given optimal alignments and their costs $(\tau_i(\rho), C_i(\rho))$ for problems $P_i$ for each $i \in [j-1]$. Some initial problems in the sequence $P_1, \ldots, P_k$ of subproblems are *base case* subproblems where the optimal alignment and its cost can be directly computed without referring to a previous subproblem. To solve a (non base case) subproblem $P_j$, we consider $V$ alternative *cases* $q : \Pi \to [V]$, i.e. $P_j$ belongs to exactly one of the $V$ cases (e.g. if $P_j = (s_1[:i'], s_2[:j'])$, we could have two cases corresponding to $s_1[i'] = s_2[j']$ and $s_1[i'] \neq s_2[j']$). Typically, $V$ will be a small constant. For any case $v = q(P_j) \in [V]$ that $P_j$ may belong to, the cost of the optimal alignment of $P_j$ is given by a minimum over $L_v$ terms of the form $c_{v,l}(\rho, P_j) = \rho \cdot w_{v,l} + \sigma_{v,l}(\rho, P_j)$, where $l \in [L_v]$, $w_{v,l} \in \mathbb{R}^d$, $\sigma_{v,l}(\rho, P_j) = C_t(\rho) \in \{C_1(\rho), \ldots, C_{j-1}(\rho)\}$ is the cost of some previously solved subproblem $P_t = (s_1[:i'_t], s_2[:j'_t]) = (s_1[:i'_{v,l,j}], s_2[:j'_{v,l,j}])$ (i.e. $t$ depends on $v, l, j$ but not on $\rho$), and $c_{v,l}(\rho, P_j)$ is the cost of alignment $\tau_{v,l}(\rho, P_j) = T_{v,l}(\tau_t(\rho))$ which extends the optimal alignment for subproblem $P_t$ by a $\rho$-independent transformation $T_{v,l}(\cdot)$. That is, the DP update for computing the cost of the optimal alignment takes the form

$$DP(\rho, P_j) = \min_l \{\rho \cdot w_{q(P_j),l} + \sigma_{q(P_j),l}(\rho, P_j)\}, \tag{1}$$

and the optimal alignment is given by $DP'(\rho, P_j) = \tau_{q(P_j),l^*}(\rho, P_j)$, where $l^* = \operatorname{argmin}_l \{\rho \cdot w_{q(P_j),l} + \sigma_{q(P_j),l}(\rho, P_j)\}$. The DP specification is completed by including base cases $\{C(s, s', \rho) = \rho \cdot w_{s,s'} \mid (s, s') \in \mathcal{B}(s_1, s_2)\}$ (or $\{\tau(s, s', \rho) = \tau_{s,s'} \mid (s, s') \in \mathcal{B}(s_1, s_2)\}$) for the

---

[9]For the sequence alignment DP in Appendix J.1.1, we have $\pi(s_1, s_2) = \Pi_{s_1, s_2}$ and we first solve the base case subproblems (which have a fixed optimal alignment for all $\rho$) followed by problems $(s_1[:i'], s_2[:j'])$ in a non-decreasing order of $i' + j'$ for any value of $\rho$.

optimal alignment DP) corresponding to a set of base case subproblems $\mathcal{B}(s_1, s_2) \subseteq \Pi_{s_1, s_2}$. Let $L = \max_{v \in [V]} L_v$ denote the maximum number of subproblems needed to compute a single DP update in any of the cases. $L$ is often small, typically 2 or 3 (see examples in Section J.1). Our main result is to provide an algorithm for computing the polytopic pieces of the dual class functions efficiently for small constants $d$ and $L$.

As indicated above, we consider the family of dynamic programs $A_\rho$ which compute the optimal alignment $\tau(s_1, s_2, \rho)$ given any pair of sequences $(s_1, s_2) \in \Sigma^m \times \Sigma^n = \Pi$ for any $\rho \in \mathbb{R}^d$. For any alignment $(t_1, t_2)$, the algorithm has a fixed real-valued utility (different from the cost function above) which captures the quality of the alignment, i.e. the utility function $u((s_1, s_2), \rho)$ only depends on the alignment $\tau(s_1, s_2, \rho)$. The dual class function is piecewise constant with convex polytopic pieces (Lemma J.5 in Appendix J.3). For any fixed problem $(s_1, s_2)$, the space of parameters $\rho$ can be partitioned into $R$ convex polytopic regions where the optimal alignment is fixed. The optimal parameter can then be found by simply comparing the costs of the alignments in each of these pieces. For the rest of this section we consider the algorithmic problem of computing these $R$ pieces efficiently.

For the clustering algorithm family, as we have seen in Section 3, we get a refinement of the parameter space with each new step (merge) performed by the algorithm. This does not hold for the sequence alignment problem. Instead we obtain the following DAG, from which the desired pieces can be obtained by looking at nodes with no out-edges (call these *terminal* nodes). Intuitively, the DAG is built by iteratively adding nodes corresponding to subproblems $P_1, \ldots, P_k$ and adding edges directed towards $P_j$ from all subproblems that appear in the DP update for it. That is, for *base case* subproblems, we have singleton nodes with no incoming edges. Using the recurrence relation (1), we note that the optimal alignment for the pair of sequences $(s_1[:i], s_2[:j])$ can be obtained from the optimal alignments for subproblems $\{(s_1[:i_{v',l}], s_2[:j_{v',l}])\}_{l \in [L_{v'}]}$ where $v' = q(s_1[:i], s_2[:j])$. The DAG for $(s_1[:i], s_2[:j])$ is therefore simply obtained by using the DAGs $G_{v',l}$ for the subproblems and adding directed edges from the terminal nodes of $G_{v',l}$ to new nodes $v_{p,i,j}$ corresponding to each piece $p$ of the partition $P[i][j]$ of $\mathcal{P}$ given by the set of pieces of $u_{(s_1[i], s_2[j])}(\rho)$. A more compact representation of the execution graph would have only a single node $v_{i,j}$ for each subproblem $(s_1[:i], s_2[:j])$ (the node stores the corresponding partition $P[i][j]$) and edges directed towards $v_{i,j}$ from nodes of subproblems used to solve $(s_1[:i], s_2[:j])$. Note that the graph depends on the problem instance $(s_1, s_2)$ as the relevant DP cases $v' = q(s_1[:i], s_2[:j])$ depend on the sequences $s_1, s_2$. A naive way to encode the execution graph would be an exponentially large tree corresponding to the recursion tree of the recurrence relation (1).

*Execution DAG.* Formally we define a *compact execution graph* $G_e = (V_e, E_e)$ as follows. For the base cases, we have nodes labeled by $(s, s') \in \mathcal{B}(s_1, s_2)$ storing the base case solutions $(w_{s,s'}, \tau_{s,s'})$ over the unpartitioned parameter space $\mathcal{P} = \mathbb{R}^d$. For $i, j > 0$, we have a node $v_{i,j}$ labeled by $(s_1[:i], s_2[:j])$ and the corresponding partition $P[i][j]$ of the parameter space, with incoming edges from nodes of the relevant subproblems $\{(s_1[:i_{v',l}], s_2[:j_{v',l}])\}_{l \in [L_{v'}]}$ where $v' = q(s_1[:i], s_2[:j])$. This graph is a DAG since every directed edge is from some node $v_{i,j}$ to a node $v_{i',j'}$ with $i' + j' > i + j$ in typical sequence alignment dynamic programs (Appendix J.1). Algorithm 5 (Appendix J.2)) gives a procedure to compute the partition of the parameter space for any given problem instance $(s_1, s_2)$ using the compact execution DAG. We give intuitive overviews of the three main routines in Algorithm 5.

- COMPUTEOVERLAYDP computes an overlay $\mathcal{P}_i$ of the input polytopic subdivisions $\{\mathcal{P}_s \mid s \in S_i\}$ and uses Clarkson's algorithm for intersecting polytopes with output-sensitive efficiency. We show that the overlay can be computed by solving at most $\tilde{R}^L$ linear programs (Algorithm 6, Lemma J.1).
- COMPUTESUBDIVISIONDP applies Algorithm 1, in each piece of the overlay we need to find the polytopic subdivision induced by $O(L^2)$ hyperplanes (the set of hyperplanes depends on the piece). This works because all relevant subproblems have the same solution within any piece of the overlay.
- Finally RESOLVEDEGENERACIESDP merges pieces where the optimal alignment is identical using a simple search over the resulting subdivision.

For our implementation of the subroutines, we have the following guarantee for Algorithm 5.

**Theorem 4.1.** *Let $R_{i,j}$ denote the number of pieces in $P[i][j]$, and $\tilde{R} = \max_{i \leq m, j \leq n} R_{i,j}$. If the time complexity for computing the optimal alignment is $O(T_{\text{DP}})$, then Algorithm 5 can be used to compute the pieces for the dual class function for any problem instance $(s_1, s_2)$, in time $O(d! L^{4d} \tilde{R}^{2L+1} T_{\text{DP}})$.*

For the special case of $d = 2$, we show that (Theorem J.6, Appendix J.3) the pieces may be computed in $O(R T_{\text{DP}})$ time using the ray search technique of [Meg78].

# 5 Acknowledgments

We thank Dan DeBlasio for useful discussions on the computational biology literature. We also thank Avrim Blum and Mikhail Khodak for helpful feedback. This material is based on work supported by the National Science Foundation under grants CCF-1910321, IIS-1901403, and SES-1919453; the Defense Advanced Research Projects Agency under cooperative agreement HR00112020003; a Simons Investigator Award; an AWS Machine Learning Research Award; an Amazon Research Award; a Bloomberg Research Grant; a Microsoft Research Faculty Fellowship.

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

# A    Additional insights and challenges

We first present an algorithm for enumerating the cells induced by a finite collection of hyperplanes in $\mathbb{R}^d$ in output-sensitive time. Our approach is based on an output-sensitive algorithm for determining the non-redundant constraints in a linear system due to Clarkson [Cla94]. At any point, we compute the *locally relevant hyperplanes* that constitute the candidate hyperplanes which could bound the cell (or piece) containing that point, using a problem-specific sub-routine, and apply Clarkson's algorithm over these to determine the facet-inducing hyperplanes for the cell as well as a collection of points in neighboring cells. This allows us to compute all the induced cells by traversing an implicit *cell adjacency graph* (Definition 2) in a breadth-first order. Our approach gives a recipe for computing the pieces of piecewise structured dual class functions with linear boundaries, which can be used to find the best parameters over a single problem instance. We further show how to compute the pieces of the sum dual class function for a collection of problem instances in output-sensitive time, by applying our approach to the collection of facet-inducing hyperplanes from each problem instance. Our approach is useful for several mechanism design problems which are known to be piecewise structured [BSV18], and we instantiate our results for two-part tariff pricing and item pricing with anonymous prices. For the single menu two-part tariff case ($d = 2$), we use additional structure of the dual class function to give a further improvement in the running time.

For linkage-based clustering, we extend the *execution tree* based approach of [BDL20] for single-parameter families to $d > 1$. The key idea is that linkage-based clustering algorithms involve a sequence of steps (called *merges*), and the algorithmic decision at any step can only refine the partition of the parameter space corresponding to a fixed sequence of steps so far. Thus, the pieces can be viewed as leaves of a tree whose nodes at depth $t$ correspond to a partition of the parameter space such that the first $t$ steps of the algorithm are identical in each piece of the partition. While in the single parameter family the pieces are intervals, we approach the significantly harder challenge of efficiently computing convex polytopic subdivisions (formally Definition 6). We essentially compute the execution tree top-down for any given problem instance, and use our hyperplane cell enumeration to compute the children nodes for any node of the tree.

We further extend the execution tree approach to a problem where all the algorithmic states corresponding to different parameter values cannot be concisely represented via a tree. For dynamic-programming based sequence alignment, we define a compact representation of the execution graph and use primitives from computational geometry, in particular for overlaying two convex subdivisions in output-polynomial time. A key challenge in this problem is that the execution graph is no longer a tree but a DAG, and we need to design an efficient representation for this DAG. We cannot directly apply Clarkson's algorithm since the number of possible alignments of two strings (and therefore the number of candidate hyperplanes) is exponential in the size of the strings. Instead, we carefully combine the polytopes of subproblems in accordance with the dynamic program update for solving the sequence alignment (for a fixed value of parameters), and the number of candidate hyperplanes is output-sensitive inside regions where all the relevant subproblems have fixed optimal alignments.

# B    Additional related work

*Data-driven algorithm design.* The framework for data-driven design was introduced by [GR16] and is surveyed in [Bal20]. It allows the design of algorithms with powerful formal guarantees when used repeatedly on inputs consisting of related problem instances, as opposed to designing and analyzing for worst-case instances. Data-driven algorithms have been proposed and analyzed for a wide variety of combinatorial problems, including clustering [BDL20], computational biology [BDD+21], mechanism design [BSV18], mixed integer linear programming [BDSV18], semi-supervised learning [BS21, SJ23], linear regression [BKST22], low-rank approximation [BIW22] and adversarially robust learning [BBSZ23]. Typically these are NP-hard problems with a variety of known heuristics. Data-driven design provides techniques to select the heuristic (from a collection or parameterized family) which is best suited for data coming from some domain. Tools for data-driven design have also been studied in the online learning setting where problems arrive in an online sequence [CAK17, BDV18, BDS20]. In this work, we focus on the statistical learning setting where the "related" problems are drawn from a fixed but unknown distribution.

*Linkage-based clustering.* [BNVW17] introduce several single parameter families to interpolate well-known linkage procedures for linkage-based clustering, and study the sample complexity of

learning the parameter. [BDL20] consider families with multiple parameters, and also consider the interpolation of different distance metrics. However the proposed execution-tree based algorithm for computing the constant-performance regions in the parameter space only applies to single-parameter families, and therefore the results can be used to interpolate linkage procedures or distance metrics, but not both. Our work removes this limitation and provides algorithms for general multi-parameter families.

Other approaches to learning a distance metric to improve the quality of clustering algorithms have been explored. For instance, [WS09] and [KPT$^+$21] consider versions of global distance metric learning, a technique to learn a linear transformation of the input space before applying a known distance metric. [SC11] also consider a hierarchical distance metric learning, which allows the learned distance function to depend on already-merged clusters. For such distance learning paradigms, the underlying objective function is often continuous, admitting gradient descent techniques. However, in our setting, the objective function is neither convex nor continuous in $\rho$; instead, it is piecewise constant for convex pieces. As a result, instead of relying on numerical techniques for finding the optimum $\rho$, our multidimensional approach enumerates all the pieces for which the objective function is constant. Furthermore, this technique determines the *exact* optimum $\rho$ rather than an approximation; one consequence of doing so is that our analysis can be extended to apply generalization guarantees from [BDL20] when learning the optimal $\rho$ over a family of instances.

*Sequence alignment.* For the problem of aligning pairs of genomic sequences, one typically obtains the best alignment using a dynamic program with some costs or weights assigned to edits of different kinds, such as insertions, substitutions, or reduplications [Wat89]. Prior work has considered learning these weights by examining the cost of alignment for the possible weight settings. [BDD$^+$21] considers the problem for how many training examples (say pairs of DNA sequences with known ancestral or evolutionary relationship) are needed to learn the best weights for good generalization on unseen examples. More closely related to our work is the line of work which proposes algorithms which compute the partition of the space of weights which result in different optimal alignments [GBN94, GS96]. In this work, we propose a new algorithm which computes this partition more efficiently.

*Pricing problems.* Data-driven algorithms have been proposed and studied for automated mechanism design [MR15, BSV18, BDV18]. Prior work on designing multi-dimensional mechanisms has focused mainly on the generalization guarantees, and include studying classes of structured mechanisms for revenue maximization [MR15, BSV16, Syr17, BSV18]. Computationally efficient algorithms have been proposed for the two-part tariff pricing problem [BPS20]. Our algorithms have output-sensitive complexity and are more efficient than [BPS20] even for worst-case outputs. Also, our techniques are more general and apply to mechanism design besides two-part tariff pricing. [BB24] consider discretization-based techniques which are not output-sensitive and known to not work in problems beyond mechanism design, for example, data-driven clustering [BNVW17].

*On cell enumeration of hyperplane arrangements.* A finite set of $t$ hyperplanes in $\mathbb{R}^d$ induces a collection of faces in dimensions $\leq d$, commonly referred to as an *arrangement*. A well-known inductive argument implies that the number of $d$-faces, or *cells*, in an arrangement of $t$ hyperplanes in $\mathbb{R}^d$ is at most $O(t^d)$. Several algorithms, ranging from topological sweep based [EG86] to incremental construction based [Xu20], can enumerate all the cells (typically represented by collections of facet-inducing hyperplanes) in optimal worst-case time $O(t^d)$. The first output-sensitive algorithm proposed for this problem was the reverse search based approach of [AF96], which runs in $O(tdR \cdot \mathrm{LP}(d, t))$ time, where $R$ denotes the output-size (number of cells in the output) and $\mathrm{LP}(d, t)$ is the time for solving a linear program in $d$ variables and $t$ constraints. This was further improved by a factor of $t$ via a more refined reverse-search [Sle99], and by additive terms via a different incremental construction based approach by [RC18]. We propose a novel approach for cell enumeration based on Clarkson's algorithm [Cla94] for removing redundant constraints from linear systems of inequalities, which asymptotically matches the worst case output-sensitive running times of these algorithms while improving over their runtime in the presence of additional structure possessed by the typical piecewise structured (Definition 1) loss functions encountered in data-driven algorithm design.

## C   Output-sensitive Parameterized Complexity

Output-sensitive algorithms have a running time that depends on the size of the output for any input problem instance. Output-sensitive analysis is frequently employed in computational geometry, for example Chan's algorithm [Cha96] computes the convex hull of a set of 2-dimensional points in time $O(n \log R)$, where $R$ is the size of the (output) convex hull. Output-sensitive algorithms are useful if the output size is variable, and 'typical' output instances are much smaller than worst case instances. Parameterized complexity extends classical complexity theory by taking into account not only the total input length $n$, but also other aspects of the problem encoded in a *parameter* $k$[10]. The motivation is to confine the super-polynomial runtime needed for solving many natural problems strictly to the parameter. Formally, a parameterized decision problem (or language) is a subset $L \subseteq \Sigma^* \times \mathbb{N}$, where $\Sigma$ is a fixed alphabet, i.e. an input $(x, k)$ to a parameterized problem consists of two parts, where the second part $k$ is the parameter. A parameterized problem $L$ is *fixed-parameter tractable* if there exists an algorithm which on a given input $(x, k) \in \Sigma^* \times \mathbb{N}$, decides whether $(x, k) \in L$ in $f(k) \cdot \text{poly}(|x|)$ time, where f is an arbitrary computable function in $k$ [DF12]. FPT is the class of all parameterized problems which are fixed-parameter tractable. In contrast, the class XP (aka *slicewise-polynomial*) consists of problems for which there is an algorithm with running time $|x|^{f(k)}$. It is known that FPT $\subsetneq$ XP. We consider an extension of the above to search problems and incorporate output-sensitivity in the following definition.

**Definition 3** (Output-polynomial Fixed Parameter Tractable). *A parameterized search problem $P : \Sigma^* \times \mathbb{N} \to \tilde{\Sigma}^*$ is said to be output-polynomial fixed-parameter tractable if there exists an algorithm which on a given input $(x, k) \in \Sigma^* \times \mathbb{N}$, computes the output $P(x, k) \in \tilde{\Sigma}^*$ in time $f(k) \cdot poly(|x|, R)$, where $R = |P(x, k)|$ is the output size and $f$ is an arbitrary computable function in $k$.*

As discussed above, output-sensitvity and fixed-parameter tractability both offer more fine-grained complexity analysis than traditional (input) polynomial time complexity. Both techniques have been employed in efficient algorithmic enumeration [Fer02, Nak04] and have gathered recent interest [FGS+19]. In this work, we consider the optimization problem of selecting tunable parameters over a continuous domain $\mathcal{C} \subset \mathbb{R}^d$ with the fixed-parameter $k = d$, the number of tunable parameters. We will design OFPT enumeration algorithms which, roughly speaking, output a finite "search space" which can be used to easily find the best parameter for the problem instance.

## D   Augmented Clarkson's algorithm

We describe here the details of the Augmented Clarkson's algorithm, which modifies the algorithm of Clarkson [Cla94] with additional bookkeeping needed for tracking the partition cells in Algorithm 1. The underlying problem solved by Clarkson's algorithm may be stated as follows.

*Problem Setup.* Given a linear system of inequalities $Ax \le b$, an inequality $A_i x \le b_i$ is said to be *redundant* in the system if the set of solutions is unchanged when the inequality is removed from the system. Given a system $(A \in \mathbb{R}^{m \times d}, b \in \mathbb{R}^m)$, find an equivalent system with no redundant inequalities.

Note that to test if a single inequality $A_i x \le b_i$ is redundant, it is sufficient to solve the following LP in $d$ variables and $m$ constraints.

$$\begin{aligned} \textbf{maximize} \quad & A_i x \\ \textbf{subject to} \quad & A_j x \le b_j, \quad \forall j \in [m] \setminus \{i\} \\ & A_i x \le b_i + 1 \end{aligned} \qquad (2)$$

Using this directly to solve the redundancy removal problem gives an algorithm with running time $m \cdot \text{LP}(d, m)$, where $\text{LP}(d, m)$ denotes the time to solve an LP in $d$ variables and $m$ constraints. This can be improved using Clarkson's algorithm if the number of non-redundant constraints $s$ is much less than the total number of constraints $m$ (Theorem 2.1).

We assume that an interior point $z \in \mathbb{R}^d$ satisfying $Ax < b$ is given. At a high level, one maintains the set of non-redundant constraints $I$ discovered so far i.e. $A_i x \le b_i$ is not redundant for each

---

[10]N.B. The term "parameter" is overloaded. It is used to refer to the real-valued parameters in the algorithm family, as well as the parameter to the optimization problem over the algorithm family.

**Algorithm 2:** AUGMENTEDCLARKSON$(z, H = (A, b), c)$

---

**Input**: $A \in \mathbb{R}^{m \times d}, b \in \mathbb{R}^m, z \in \mathbb{R}^d$, sign-pattern $c \in \{0, 1, -1\}^m$.
**Output**: list of non-redundant hyperplanes $I \subseteq [m]$, points in neighboring cells $Z \subset \mathbb{R}^d$.
$I \leftarrow \emptyset, J \leftarrow [m], Z \leftarrow \emptyset$
**while** $J \neq \emptyset$ **do**
> Select $k \in J$
> Detect if $A_k x \leq b_k$ is redundant in $A_{I \cup \{k\}} x \leq b_{I \cup \{k\}}$ by solving LP (2).
> $x^* \leftarrow$ optimal solution of the above LP
> **if** *redundant* **then**
> > $J \leftarrow J \setminus \{k\}$
>
> $j, z^* \leftarrow$ RayShoot$(A, b, z, x^*)$;  /* Computes $j$ such that $A_j x = b_j$ is a
> facet-inducing hyperplane hit by ray from $z$ along $x^* - z$ */
> $J \leftarrow J \setminus \{j\}$
> $c'_j \leftarrow -c_j, c'_i \leftarrow c_i \; \forall i \in [m] \setminus \{j\}$
> $I \leftarrow I \cup \{j\}, Z \leftarrow Z \cup \{[c', z^*]\}$

**return** $I, Z$

---

$i \in I$. When testing a new index $k$, the algorithm solves an LP of the form 2 and either detects that $A_k x \leq b_k$ is redundant, or finds index $j \in [m] \setminus I$ such that $A_j x \leq b_j$ is non-redundant. The latter involves the use of a procedure RayShoot$(A, b, z, x)$ which finds the non-redundant hyperplane hit by a ray originating from $z$ in the direction $x - z$ ($x \in \mathbb{R}^d$) in the system $A, b$. The size of the LP needed for this test is LP$(\mathrm{d}, |\mathrm{I}| + 1)$ from which the complexity follows.

To implement the RayShoot procedure, we can simply find the intersections of the ray $x^* - z$ with the hyperplanes $A_j x \leq b_j$ and output the one closest to $z$ (defining the cell facet in that direction). We also output an interior point from the adjacent cell during this computation, which saves us time relative to [Sle99] where the interior point is computed for each cell (our Clarkson based approach also circumvents their need for the Raindrop procedure [BFM97]). Finally we state the running time guarantee of Algorithm 2, which follows from the original result of Clarkson [Cla94].

---

**Algorithm 3:** RayShoot

---

**Input**: $A \in \mathbb{R}^{m \times d}, b \in \mathbb{R}^m, z \in \mathbb{R}^d, x \in \mathbb{R}^d$.
**if** $A_i \cdot (x - z) = 0$ *for some* $i$ **then**
> $z \leftarrow z + (\epsilon, \epsilon^2, \ldots, \epsilon^d)$ for sufficiently small $\epsilon$ ;  /* Ensure general position. */

$t_i \leftarrow \frac{b_i - A_i \cdot z}{A_i \cdot (x - z)}$;  /* intersection of $z + t(x - z)$ with $A_i x = b_i$ */
$j = \mathrm{argmin}_i \{t_i \mid t_i > 0\}, t' = \max\{\min_{i \neq j}\{t_i \mid t_i > 0\}, 0\}$
**return** $j, z + \frac{t_j + t'}{2}(x - z)$

---

**Theorem D.1** (Augmented Clarkson's algorithm). *Given a list $L$ of $k$ half-space constraints in $d$ dimensions, Algorithm 2 outputs the set $I \subseteq [L]$ of non-redundant constraints in $L$, as well as auxiliary neighbor information, in time $O(k \cdot \mathrm{LP}(d, |I| + 1))$, where $\mathrm{LP}(v, c)$ is the time for solving an LP with $v$ variables and $c$ constraints.*

## E  Additional details and proofs from Section 2

*Proof of Theorem 2.2.* Algorithm 1 maintains a set of *visited* or *explored* cells in $C$, and their bounding hyperplanes (corresponding to cell facets) in $\tilde{C}$. It also maintains a queue of cells, such that each cell in the queue has been discovered in Line 12 as a neighbor of some visited cell in $C$, but is yet to be explored itself. The algorithm detects if a cell has been visited before by using its sign pattern on the hyperplanes in $H$. This can be done in $O(d \log t)$ time, since $|C| = O(t^d)$ using a well-known combinatorial fact (e.g. [Buc43]). For a new cell $c$, we run AUGMENTEDCLARKSON to compute its bounding hyperplanes $I_c$ as well as sign patterns for neighbors in time $O(t_{\mathrm{LRS}} \cdot \mathrm{LP}(d, |I_c| + 1))$ by Theorem 2.1. The computed neighbors are added to the queue in Line 14. A cell $c$ gets added to the queue at this step up to $|I_c|$ times, but is not explored if already visited. Thus we run up to $V$ iterations

of AUGMENTEDCLARKSON and up to $1 + \sum_{c \in V_H} |I_c| = 2E + 1$ queue insertions/removals. Using efficient union-find data-structures, the set union and membership queries (for $C$ and $\tilde{C}$) can be done in $\tilde{O}(1)$ time per iteration of the loop. So total time over the $V$ cell explorations and no more than $2E + 1$ iterations of the **while** loop is $\tilde{O}(dE) + O(t_{\text{LRS}} \cdot \sum_{c \in V_H} \text{LP}(d, |I_c| + 1)) + O(VT_{\text{CLRS}})$. $\quad \square$

*Proof of Theorem 2.3.* We will apply Algorithm 1 and compute the locally relevant hyperplanes by simply taking a union of the facet-inducing hyperplanes at any point $\mathbf{x}$, across the problem instances.

Let $G^{(i)} = (V^{(i)}, E^{(i)})$ denote the cell adjacency graph for the cells in $\tilde{C}_i$. We apply Algorithm 1 implicitly over $H = \cup_i \overline{H}^{(i)}$ where $\overline{H}^{(i)}$ is the collection of facet-inducing hyperplanes in $\tilde{C}_i$. To implement COMPUTELOCALLYRELEVANTSEPARATORS$(\mathbf{x}, H)$ we simply search the computed partition cell $\tilde{C}_i$ for the cell containing $\mathbf{x}$ in each problem instance $x_i$ in time $O(\sum_i |E^{(i)}|) = O(m|E_S|)$, obtain the set of corresponding facet-inducing hyperplanes $H_{\mathbf{x}}^{(i)}$, and output $\tilde{H}_{\mathbf{x}} = \cup_i H_{\mathbf{x}}^{(i)}$ in time $O(mt_{\text{LRS}})$. The former step only needs to be computed once, as the partition cells for subsequent points can be tracked in $O(m)$ time. Theorem 2.2 now gives a running time of $\tilde{O}(d|E_S| + m|E_S| + mt_{\text{LRS}}|V_S| + mt_{\text{LRS}} \cdot \sum_{c \in V_S} \text{LP}(d, |I_c| + 1))$. $\quad \square$

# F  Profit maximization in pricing problems

Prior work [MR15, BSV18] on data-driven mechanism design has shown that the profit as a function of the prices (parameters) is $(\mathcal{F}, t)$-decomposable with $\mathcal{F}$ the set of linear functions on $\mathbb{R}^d$ for a large number of mechanism classes (named $(d, t)$-delineable mechanisms in [BSV18]). We will instantiate our approach for multi-item pricing problems which are $(\mathcal{F}, t)$-decomposable and analyse the running times. In contrast, recent work [BB24] employs data-independent discretization for computationally efficient data-driven algorithm design for mechanism design problems even in the worst-case. This discretization based approach is not output-sensitive and is known to not work well for other applications like data-driven clustering.

## F.1  Two-part tariff pricing

In the *two-part tariff* problem [Lew41, Oi71], the seller with multiple identical items charges a fixed price, as well as a price per item purchased. For example, cab meters often charge a base cost for any trip and an additional cost per mile traveled. Subscription or membership programs often require an upfront joining fee plus a membership fee per renewal period, or per service usage. Often there is a menu or tier of prices, i.e. a company may design multiple subscription levels (say basic, silver, gold, platinum), each more expensive than the previous but providing a cheaper per-unit price. Given access to market data (i.e. profits for different pricing schemes for typical buyers) we would like to learn how to set the base and per-item prices to maximize the profit. We define these settings formally as follows.

*Two-part tariffs.* The seller has $K$ identical units of an item. Suppose the buyers have valuation functions $v_i : \{1, \ldots, K\} \to \mathbb{R}_{\geq 0}$ where $i \in \{1, \ldots, m\}$ denotes the buyer, and the value is assumed to be zero if no items are bought. Buyer $i$ will purchase $q$ quantities of the item that maximizes their utility $u_i(q) = v_i(q) - (p_1 + p_2 q)$, buying zero units if the utility is negative for each $q > 0$. The revenue, which we want to maximize as the seller, is zero if no item is bought, and $p_1 + p_2 q$ if $q > 0$ items are bought. The algorithmic parameter we want to select is the price $\rho = \langle p_1, p_2 \rangle$, and the problem instances are specified by the valuations $v_i$. We also consider a generalization of the above scheme: instead of just a single two-part tariff (TPT), suppose the seller provides a menu of TPTs $(p_1^1, p_2^1), \ldots, (p_1^\ell, p_2^\ell)$ of length $\ell$. Buyer $i$ selects a tariff $(p_1^j, p_2^j)$ from the menu as well as the item quantity $q$ to maximize their utility $u_i^j(q) = v_i(q) - (p_1^j + p_2^j q)$. This problem has $2\ell$ parameters, $\rho = (p_1^1, p_2^1, \ldots, p_1^\ell, p_2^\ell)$, and the single two-part tariff setting corresponds to $\ell = 1$.

The dual class functions in this case are known to be piecewise linear with linear boundaries ([BSV18], restated as Lemma G.2 in Appendix G). We will now implement Algorithm 1 for this problem by specifying how to compute the locally relevant hyperplanes. For any price vector $\mathbf{x} = \rho$, say the buyers buy quantities $(q_1, \ldots, q_m) \in \{0, \ldots, K\}^m$ according to tariffs $(j_1, \ldots, j_m) \in [\ell]^m$. For a fixed price vector this can be done in time $O(mK\ell)$ by computing $\text{argmax}_{q,j} \, u_i^j(q)$ for each buyer

at that price for each two-part tariff in the menu. Then for each buyer we have $K(\ell - 1)$ potential alternative quantities and tariffs given by hyperplanes $u_i^{j_i}(q_i) \geq u_i^{j'}(q'), q' \neq q_i, j' \neq j_i$, for a total of $t_{\text{LRS}} = mK(\ell - 1)$ locally relevant hyperplanes. Thus $T_{\text{CLRS}} = O(mK\ell)$ for the above approach, and Theorem 2.2 implies the following runtime bound.

**Theorem F.1.** *There exists an implementation of* COMPUTELOCALLYRELEVANTSEPARATORS *in Algorithm 1, which given valuation function $v(\cdot)$ for a single problem instance, computes all the $R$ pieces of the dual class function $u_v(\cdot)$ in time $\tilde{O}(R^2(2\ell)^{\ell+2}K)$, where the menu length is $\ell$, and there are $K$ units of the good.*

Theorem F.1 together with Theorem 2.3 implies an implementation of the search-ERM problem over $m$ buyers (with valuation functions $v_i(\cdot)$ for $i \in [m]$) in time $O(R_\Sigma^2(2\ell)^{\ell+2}mK)$, where $R_\Sigma$ denotes the number of pieces in the total dual class function $U_{\langle v_1,\ldots,v_m \rangle}(\cdot) = \sum_i u_{v_i}(\cdot)$ (formally, Corollary G.1 in Appendix G). In contrast, prior work for this problem has only obtained an XP runtime of $(mK)^{O(\ell)}$ [BPS20]. For the special case $\ell = 1$, we also provide an algorithm (Algorithm 4 in Appendix G.2) that uses additional structure of the polytopes and employs a computational geometry algorithm due to [CE92] to compute the pieces in optimal $O(mK \log(mK) + R_\Sigma)$ time, improving over the previously best known runtime of $O(m^3 K^3)$ due to [BPS20] even for worst-case $R_\Sigma$. The worst-case improvement follows from a bound of $R_\Sigma = O(m^2 K)$ on the number of pieces, which we establish in Theorem G.3. We further show that our running time for $\ell = 1$ is asymptotically optimal under the algebraic decision-tree model of computation, by a linear time reduction from the element uniqueness problem (Theorem G.6).

### F.2  Item-Pricing with anonymous prices

We will consider a market with a single seller, interested in designing a mechanism to sell $m$ distinct items to $n$ buyers. We represent a *bundle* of items by a quantity vector $q \in \mathbb{Z}_{\geq 0}^m$, such that the number of units of the $i$th item in the bundle denoted by $q$ is given by its $i$th component $q[i]$. In particular, the bundle consisting of a single copy of item $i$ is denoted by the standard basis vector $e_i$, where $e_i[j] = \mathbb{I}\{i = j\}$, where $\mathbb{I}\{\cdot\}$ is the 0-1 valued indicator function. Each buyer $j \in [n]$ has a valuation function $v_j : \mathbb{Z}_{\geq 0}^m \to \mathbb{R}_{\geq 0}$ over bundles of items. We denote an allocation as $Q = (q_1, \ldots, q_n)$ where $q_j$ is the bundle of items that buyer $j$ receives under allocation $Q$. Under *anonymous* prices, the seller sets a price $p_i$ per item $i$. There is some fixed but arbitrary ordering on the buyers such that the first buyer in the ordering arrives first and buys the bundle of items that maximizes their utility, then the next buyer in the ordering arrives, and so on. For a given buyer $j$ and bundle pair $q_1, q_2 \in \{0, 1\}^m$, buyer $j$ will prefer bundle $q_1$ over bundle $q_2$ so long as $u_j(q_1) > u_j(q_2)$ where $u_j(q) = v_j(q) - \sum_{i:q[i]=1} p_i$. Therefore, for a fixed set of buyer values and for each buyer $j$, their preference ordering over the bundles is completely determined by the $\binom{2^m}{2}$ hyperplanes. The dual class functions are known to be piecewise linear with linear boundaries [BSV18].

To implement Algorithm 1 for this problem we specify how to compute the locally relevant hyperplanes. For any price vector $\mathbf{x} = (p_1, \ldots, p_m)$, say the buyers buy bundles $(q_1, \ldots, q_n)$. For a fixed price vector this can be done in time $O(n2^m)$ by computing $\arg\max_{q \subseteq I_j} u_j(q)$ for each buyer at that price vector, where $I_j$ denotes the remaining items after allocations to buyers $1, \ldots, j-1$ at that price. Then for each buyer we have at most $2^m - 1$ potential alternative bundles given by hyperplanes $u_j(q) \geq u_j(q'), q' \neq q_i$, for a total of $t_{\text{LRS}} \leq n(2^m - 1)$ locally relevant hyperplanes. Thus $T_{\text{CLRS}} = O(n2^m)$ for the above approach, and Theorem 2.2 implies the following runtime bound.

**Theorem F.2.** *There exists an implementation of* COMPUTELOCALLYRELEVANTSEPARATORS *in Algorithm 1, which given valuation functions $v_i(\cdot)$ for $i \in [n]$, computes all the $R$ pieces of the dual class function in time $\tilde{O}(R^2(2m)^m n)$, where there are $m$ items and $n$ buyers.*

*Proof.* In the terminology of Theorem 2.2, we have $d = m$, $E \leq R^2$, $V = R$, $T_{\text{CLRS}} = O(n2^m)$, $t_{\text{LRS}} = n(2^m - 1)$. By [Cha18], we have $\sum_{c \in V_H} \text{LP}(d, |I_c| + 1) \leq O(Ed^d) \leq O(R^2 m^m)$. Thus, Theorem 2.2 implies a runtime bound on Algorithm 1 of $\tilde{O}(dE + VT_{\text{CLRS}} + t_{\text{LRS}} \cdot \sum_{c \in V_H} \text{LP}(d, |I_c| + 1)) = \tilde{O}(R^2(2m)^m n)$. $\qquad\square$

Our approach yields an efficient algorithm when the number of items $m$ and the number of dual class function pieces $R$ are small. Prior work on item-pricing with anonymous prices has only focussed on sample complexity of the data-driven design problem [BSV18].

## G  Additional details and proofs from Section F

*Proof of Theorem F.1.* In the terminology of Theorem 2.2, we have $d = 2\ell$, $E \leq R^2$, $V = R$, $T_{\text{CLRS}} = O(K\ell)$, $t_{\text{LRS}} \leq K\ell$. By [Cha18], we have $\sum_{c \in V_H} \text{LP}(d, |I_c| + 1) \leq O(Ed^{d/2+1}) \leq O(R^2(2\ell)^{\ell+1})$. Thus, Theorem 2.2 implies a runtime bound on Algorithm 1 of $\tilde{O}(dE + VT_{\text{CLRS}} + t_{\text{LRS}} \cdot \sum_{c \in V_H} \text{LP}(d, |I_c| + 1)) = \tilde{O}(R^2(2\ell)^{\ell+2}K)$. $\qquad\square$

**Corollary G.1.** *There exists an implementation of* COMPUTELOCALLYRELEVANTSEPARATORS *in Algorithm 1, which given valuation functions $v_i(\cdot)$ for $i \in [m]$, computes all the $R_\Sigma$ pieces of the total dual class function $U_{\langle v_1,\ldots,v_m \rangle}(\cdot) = \sum_i u_{v_i}(\cdot)$ in time $\tilde{O}(R_\Sigma^2(2\ell)^{\ell+2}mK)$, where the menu length is $\ell$, there are $K$ units of the good and $m$ is the number of buyers.*

*Proof.* We first compute the pieces for each of the problem instances (single buyers) and then the pieces in the sum dual class function using Theorem 2.3. By Theorem F.1, the former takes time $\tilde{O}(mR^2(2\ell)^{\ell+2}K)$ and the latter can be implemented in time $O((m + 2\ell)R_\Sigma^2 + mK\ell \sum_{c \in V_S} \text{LP}(d, |I_c| + 1)) = \tilde{O}(R_\Sigma^2(2\ell)^{\ell+2}mK)$, which dominates the overall running time. $\quad\square$

### G.1  Piecewise structure of the dual class function

The following lemma restates the result from [BSV18] in terms of Definition 1. Note that $u_\rho$ in the following denotes the revenue function (or seller's utility) and should not be confused with the buyer utility function $u_i$.

**Lemma G.2.** *Let $\mathcal{U}$ be the set of functions $\{u_\rho : v(\cdot) \mapsto p_1^{j^*} + p_2^{j^*} q^* \mid q^*, j^* = \text{argmax}_{q,j} v(q) - \rho^j \cdot \langle 1, q \rangle, \rho^j = \langle p_1^j, p_2^j \rangle\}$ that map valuations $v(\cdot)$ to $\mathbb{R}$. The dual class $\mathcal{U}^*$ is $(\mathcal{F}, (K\ell)^2)$-piecewise decomposable, where $\mathcal{F} = \{f_c : \mathcal{U} \to \mathbb{R} \mid c \in \mathbb{R}^{2\ell}\}$ consists of linear functions $f_c : u_\rho \mapsto \rho \cdot c$.*

We also bound the number of pieces $R$ in the worst-case for $\ell = 1$. The following bound implies that our algorithm is better than prior best algorithm which achieves an $O(m^3K^3)$ runtime bound, even for worst case outputs.

**Theorem G.3.** *Let menu length $\ell = 1$. The number of pieces $R_\Sigma$ in the total dual class function $U_{\langle v_1,\ldots,v_m \rangle}(\cdot) = \sum_i u_{v_i}(\cdot)$ is at most $O(m^2K)$.*

*Proof.* By Lemma G.2, the dual class is $(\mathcal{F}, K^2)$-piecewise decomposable, where $\mathcal{F}$ is the class of linear functions. That is, the two-dimensional parameter space $(p_1, p_2)$ can be partitioned into polygons by at most $K^2$ straight lines such that any dual class function $u_{v_i}$ is a linear function inside any polygon.

We first tighten the above result to show that the dual class is in fact $(\mathcal{F}, 2K + 2)$-piecewise decomposable, that is the number of bounding lines for the pieces is $O(K)$. This seems counterintuitive since for any buyer $i$, we have $\Theta(K^2)$ lines $u_i(q) \geq u_i(q')$ for $q < q' \in \{0, \ldots, K\}$. If $q > 0$, $u_i(q) = u_i(q')$ are axis-parallel lines with intercepts $\frac{v_i(q')-v_i(q)}{q'-q}$. Since for any pair $q, q'$ the buyer has a fixed (but opposite) preference on either side of the axis-parallel line, we have at most $K$ distinct horizontal 'slabs' corresponding to buyer's preference of quantities, i.e. regions between lines $p_2 = a$ and $p_2 = b$ for some $a, b > 0$. Thus we have at most $K$ non-redundant lines. Together with another $K$ lines $u_i(0) = u_i(q')$ and the axes, we have $2K + 2$ bounding lines in all as claimed.

We will next use this tighter result to bound the number of points of intersection of non-collinear non-axial bounding lines, let's call them *crossing points*, across all instances $\langle v_1, \ldots, v_m \rangle$. Consider a pair of instances given by buyer valuation functions $v_i, v_j$. We will establish that the number of crossing points are at most $4K$ for the pair of instances. Let $l_i$ and $l_j$ be bounding lines for the pieces of $u_{v_i}$ and $u_{v_j}$ respectively. If they are both axis-parallel, then they cannot result in a crossing point. For pairs of bounding lines $l_i$ and $l_j$ such that $l_i$ is axis-parallel and $l_j$ is not, we can have at most $K$ crossing points in all. This is because any fixed $l_i$ can intersect at most one such $l_j$ since buyer $j$'s

preferred quantity $q_j$ is fixed along any horizontal line, unless $q_j$ changes across $l_i$ in which case the crossing points for the consecutive $l_j$'s coincide. Thus, there is at most one such crossing point for each of at most $K$ axis-parallel $l_i$'s. By symmetry, there are at most $K$ crossing points between $l_i, l_j$ where $l_j$ is axis parallel and $l_i$ is not. Finally, if neither $l_i, l_j$ is axis parallel, we claim there can be no more than $2K$ crossing points. Indeed, if we arrange these points in the order of increasing $p_2$, then the preferred quantity of at least one of the buyers $i$ or $j$ strictly decreases between consecutive crossing points. Thus, across all instances, there are at most $2m^2 K$ crossing points.

Finally, observe that the cell adjacency graph $G_U$ for the pieces of the total dual class function $U_{\langle v_1,...,v_m \rangle}(\cdot)$ is planar in this case. The vertices of this graph correspond to crossing points, or intersections of bounding lines with the axes. The latter is clearly $O(mK)$ since there are $O(K)$ bounding lines in any problem instance. Using the above bound on crossing points, the number of vertices in $G_U$ is $O(m^2 K)$. Since $G_U$ is a simple, connected, planar graph, the number of faces is no more than twice the number of vertices and therefore the number of pieces $R_\Sigma$ is also $O(m^2 K)$. $\quad\square$

### G.2  Optimal algorithm for Single TPT pricing

Consider the setting with menu-length $\ell = 1$. The key insight is to characterize the polytopic structure of the pieces of the dual class function for a single buyer. We do this in Lemma G.4.

**Lemma G.4.** *Consider a single buyer with valuation function $v(\cdot)$. The buyer buys zero units of the item except for a set $\varrho_v \subset \mathbb{R}^2$, where $\varrho_v$ is a convex polygon with at most $K + 2$ sides. Moreover, $\varrho_v$ can be subdivided into $K' \leq K$ polygons $\varrho_v^{(i)}$, each a triangle or a trapezoid with bases parallel to the $\rho_1$-axis, such that for each $i \in [K']$ the buyer buys the same quantity $q^{(i)}$ of the item for all prices in $\varrho_v^{(i)}$.*

*Proof.* We proceed by an induction on $K$, the number of items. For $K = 1$, it is straightforward to verify that $\varrho_v$ is the triangle $p_1 \geq 0, p_2 \geq 0, p_1 + p_2 \leq v(1)$.

Let $K > 1$. If we consider the restriction of the valuation function $v(\cdot)$ to $K - 1$ items, we have a convex polygon $\varrho'_v$ satisfying the induction hypothesis. To account for the $K$-th item we only need to consider the region $p_1 \geq 0, p_2 \geq 0, p_2 \leq \frac{v(K)-v(q)}{K-q}$ for $0 < q < K$, and $p_1 + p_2 K \leq v(K)$. If this region is empty, $\varrho_v = \varrho'_v$, and we are done. Otherwise, denoted by $\varrho_v^{(K')}$ with $q^{(K')} = K$, the region where the buyer would buy $K$ units of the item is a trapezoid with bases parallel to the $\rho_1$-axis. We claim that $\varrho_v = \left( \varrho'_v \cap p_2 \leq \frac{v(K)-v(q)}{K-q} \right) \cup \varrho_v^{(K')}$ and it satisfies the properties in the lemma.

We have $q^{(K'-1)} = \operatorname{argmin}_q \frac{v(K)-v(q)}{K-q}$ such that the buyer's preference changes from $q' = q^{(K'-1)}$ to $q^{(K')} = K$ units across the line $p_2 = \frac{v(K)-v(q')}{K-q'}$. To prove $\varrho_v$ is convex, we use the inductive hypothesis on $\varrho'_v$ and observe that $\rho_1 = v(K) - p_2 K$ coincides with $\rho'_1 = v(q') - p_2 q'$ for $p_2 = \frac{v(K)-v(q')}{K-q'}$. Also the only side of $\varrho_v$ that is not present in $\varrho'_v$ lies along the line $p_1 + p_2 K = v(K)$, thus $\varrho_v$ has at most $K+2$ sides. The subdivison property is also readily verified given the construction of $\varrho_v$ from $\varrho'_v$ and $\varrho_v^{(K')}$. $\quad\square$

Based on this structure, we propose Algorithm 4 which runs in in $O(mK \log(mK) + R_\Sigma)$ time.

**Theorem G.5.** *There is an algorithm (Algorithm 4) that, given valuation functions $v_i(\cdot)$ for $i \in [m]$, computes all the $R$ pieces of the total dual class function $U_{\langle v_1,...,v_m \rangle}(\cdot)$ for $K$ units of the good from the $m$ samples in $O(mK \log(mK) + R_\Sigma)$ time.*

*Proof.* Note that if $0 < q < q'$ and $v_i(q) > v_i(q')$, then for any $\rho_1 \geq 0, \rho_2 \geq 0$ we have that $u_i(q) = v_i(q) - (p_1 + p_2 q) > v_i(q') - (p_1 + p_2 q)$, or the buyer will never prefer quantity $q'$ of the item over the entire tariff domain. Thus, we will assume the valuations $v_i(q)$ are monotonic in $q$ (we can simply ignore valuations at the larger value for any violation). Algorithm 4 exploits the structure in Lemma G.4 and computes the $O(K)$ line segments bounding the dual class pieces for a single buyer $i$ in $O(K)$ time. Across $m$ buyers, we have $O(mK)$ line segments (computed in $O(mK)$ time). The topological plane-sweep based algorithm of [CE92] now computes all the intersection points in $O(mK(\log mK) + R_\Sigma)$ time. Here we have used that the number of polytopic vertices is $O(R_\Sigma)$ using standard result for planar graphs. $\quad\square$

**Algorithm 4:** ComputeFixedAllocationRegions

---

**Input**: $v_i(\cdot)$, valuation functions
1. For $i = 1 \ldots m$ do
2. $Q \leftarrow \emptyset$ (stack), $q' = 1$, $h = v_i(1)$.
3. For $q = 2 \ldots K$ do
3.1     $h' \leftarrow (v_i(q) - v_i(q'))/(q - q')$
3.2     if $0 < h' < h$
         $h \leftarrow h'$
         Push $(q, h')$ onto $Q$
         $q' \leftarrow q$
3.3     else if $0 < h'$
         while $h' \geq h$ do
             Pop $(q', h)$ from $Q$
             $(q_1, h_1) \leftarrow \text{Top}(Q)$
             $h' \leftarrow (v_i(q) - v_i(q_1))/(q - q_1)$
             $h \leftarrow h_1$
             $q' \leftarrow q$
         Push $(q, h')$ onto $Q$
4. For $(q, h) \in Q$ do
         Obtain $\varrho_{v_i}^{(j)}$ for $q^{(j)} = q$ using lines $p_2 = h$ and $p_1 = v(q) - p_2 q$.
5. Compute intersection of segments in $\varrho_{v_i}^{(j)}$ for $i \in [m]$ using [CE92].
6. Use the intersection points to compute boundaries of all polygons (pieces) formed by the
  intersections.

---

We further show that this bound is essentially optimal. A runtime lower bound of $\Omega(mK + R_\Sigma)$ follows simply due to the amount of time needed for reading the complete input and producing the complete output. We prove a stronger lower bound which matches the above upper bound by reduction to the *element uniqueness problem* (given a list of $n$ numbers, are there any duplicates?) for which an $\Omega(n \log n)$ lower bound is known in the algebraic decision-tree model of computation.

**Theorem G.6.** *Given a list of $n$ numbers $\mathcal{L} = \langle x_1, \ldots, x_n \rangle \in \mathbb{N}^n$, there is a linear time reduction to a $m$-buyer, $K$-item TPT pricing given by $v_i(\cdot), i \in [m]$, with $mK = \Theta(n)$, such that the pieces of the total dual class function $U_{\langle v_1, \ldots, v_m \rangle}(\cdot)$ can be used to solve the element uniqueness problem for $\mathcal{L}$ in $O(n)$ time.*

*Proof.* Let $mK = n$ be any factorization of $n$ into two factors. We construct a $m$-buyer, $K + 1$ item single TPT pricing scheme as follows. Define $y_j = x_j + \max_k x_k + 1$ for each $x_j$ in the list $\mathcal{L}$. For every buyer $i \in [m]$, we set $v_i(1) = \max_k x_k + 1$ and $v_i(q + 1) = \sum_{j=1}^{q} y_{j+(i-1)K}$ for each $q \in [K]$. Buyer $i$'s pieces include the segments $p_2 = (v_i(q + 1) - v_i(q))/(q + 1 - 1) = x_{q+(i-1)K}$ for $q \in [K]$ (Lemma G.4). Thus, across all buyers $i \in [m]$, we have $mK = n$ segments along the lines $p_2 = x_j$ for $j \in [n]$. We say a duplicate is present if there are fewer than $mK$ segments parallel to the $p_1$-axis in the pieces of the total dual class function, otherwise we say 'No' (i.e. all elements are distinct). This completes the linear-time reduction. $\square$

## H   Comparing the quality of single, complete and median linkage procedures on different data distributions

We will construct clutering instances where each of two-point based linkage procedures, i.e. single, complete and median linkage, dominates the other two procedures. Let $T_{\text{sgl}}^S$, $T_{\text{cmpl}}^S$ and $T_{\text{med}}^S$ denote the cluster tree on clustering instance $S$ using $D_{\text{sgl}}$, $D_{\text{cmpl}}$ and $D_{\text{med}}$ as the merge function (defined in Section 3) respectively for some distance metric $d$ which will be evident from context. We have the following theorem.

**Theorem H.1.** *For any $n \geq 10$, for $i \in \{1, 2, 3\}$, there exist clustering instances $S_i$ with $|S_i| = n$ and target clusterings $\mathcal{C}_i$ such that the hamming loss of the optimal pruning of the cluster trees constructed using single, complete and median linkage procedures (using the same distance metric $d$) satisfy*

(i) $\ell(T_{sgl}^{S_1}, \mathcal{C}_1) = O(\frac{1}{n})$, $\ell(T_{cmpl}^{S_1}, \mathcal{C}_1) = \Omega(1)$ and $\ell(T_{med}^{S_1}, \mathcal{C}_1) = \Omega(1)$,

(ii) $\ell(T_{cmpl}^{S_2}, \mathcal{C}_2) = O(\frac{1}{n})$, $\ell(T_{sgl}^{S_2}, \mathcal{C}_2) = \Omega(1)$ and $\ell(T_{med}^{S_2}, \mathcal{C}_2) = \Omega(1)$,

(iii) $\ell(T_{med}^{S_3}, \mathcal{C}_3) = O(\frac{1}{n})$, $\ell(T_{cmpl}^{S_3}, \mathcal{C}_3) = \Omega(1)$ and $\ell(T_{sgl}^{S_3}, \mathcal{C}_3) = \Omega(1)$.

*Proof.* In the following constructions we will have $S_i \subset \mathbb{R}^2$ and the distance metric $d$ will be the Euclidean metric. Also we will have number of target clusters $k = 2$.

**Construction of $S_1, \mathcal{C}_1$.** For $S_1$, we will specify the points using their polar coordinates. We place a single point $x$ at the origin $(0, \phi)$ and $\frac{n-1}{8}$ points each along the unit circle at locations $y_1 = (1, 0), y_2 = (1, \frac{\pi}{4} - \epsilon), y_3 = (1, \frac{\pi}{2}), y_4 = (1, \frac{3\pi}{4} - \epsilon), y_5 = (1, \pi), y_6 = (1, \frac{5\pi}{4} - \epsilon), y_7 = (1, \frac{3\pi}{2})$ and $y_8 = (1, \frac{7\pi}{4} - \epsilon)$, where $\epsilon = 0.001$. Also set $\mathcal{C}_1 = \{\{x\}, S_1 \setminus \{x\}\}$.

In each linkage procedure, the first $n - 9$ merges will join coincident points at locations $y_j, j \in [8]$, let $\tilde{y}_j$ denote the corresponding sets of merged points. The next four merges will be $z_j := \{\tilde{y}_j, \tilde{y}_{j+1}\}$ for $j \in \{1, 3, 5, 7\}$ for each procedure since $d(y_j, y_{j+1}) = \sqrt{2 - 2\cos(\frac{\pi}{4} - \epsilon)} < \min\{\sqrt{2 - 2\cos(\frac{\pi}{4} + \epsilon)}, 1\}$, again common across all procedures. Now single linkage will continue to merge clusters on the unit circle since $\sqrt{2 - 2\cos(\frac{\pi}{4} + \epsilon)} < 1$, however both complete and median linkage will join each of $z_j, j \in \{1, 3, 5, 7\}$ to the singleton cluster $\{x\}$ since the median (and therefore also the maximum distance between points in $z_j, z_{j'}, j \neq j'$ is at least $\sqrt{2} > 1$. Therefore a 2-pruning[11] of $T_{sgl}^{S_1}$ yields $\mathcal{C}_1$ i.e $\ell(T_{sgl}^{S_1}, \mathcal{C}_1) = 0$, while a 2-pruning of $T_{cmpl}^{S_1}$ or $T_{med}^{S_1}$ would yield $\{z_j, S_1 \setminus z_j\}$ for some $j \in \{1, 3, 5, 7\}$, corresponding to a hamming loss of $\frac{1}{4} + \Omega(\frac{1}{n}) = \Omega(1)$.

**Construction of $S_2, \mathcal{C}_2$.** For $S_2$, we will specify the points using their Cartesian coordinates. We place single points $x_1, x_2$ at $(0, 0)$ and $(3.2, 0.5)$ and $\frac{n-2}{2}$ points each at $y_1 = (1.1, 1.8)$ and $y_2 = (1.8, 0.5)$. We set $\mathcal{C}_2 = \{\{(x, y) \in S_2 \mid y > 1\}, \{(x, y) \in \check{S}_2 \mid y \leq 1\}\}$. The distances between pairs of points may be ordered as

$$d(x_2, y_2) = 1.4 < d(y_1, y_2) \approx 1.5 < d(x_1, y_2) \approx 1.9 < d(x_1, y_1) \approx 2.1 < d(x_2, y_1) \approx 2.5$$
$$< d(x_1, x_2)$$

All linkage procedures will merge the coincident points at $y_1$ and $y_2$ (respectively) for the first $n - 4$ merges. Denote the corresponding clusters by $\tilde{y}_1$ and $\tilde{y}_2$ respectively. The next merge will be $z_2 := \{x_2, \tilde{y}_2\}$ in all cases. Now single linkage will join $z_2$ with $\tilde{y}_1$. Further, since $n \geq 10$, $\frac{n-2}{2} \geq 4$ and therefore the median distance between $z_2$ and $\tilde{y}_1$ is also $d(y_1, y_2)$. However, since $d(x_1, y_1) < d(x_2, y_1)$, the complete linkage procedure will merge $\{x_1, z_2\}$. Finally, the two remaining clusters are merged in each of the two procedures. Clearly, 2-pruning of $T_{cmpl}^{S_2}$ yields $\mathcal{C}_2$ or $\ell(T_{cmpl}^{S_2}, \mathcal{C}_2) = 0$. However, $\ell(T_{sgl}^{S_2}, \mathcal{C}_2) = \ell(T_{med}^{S_2}, \mathcal{C}_2) = \frac{1}{2} - O(\frac{1}{n}) = \Omega(1)$.

**Construction of $S_3, \mathcal{C}_3$.** We specify the points in $S_3$ using their Cartesian coordinates. We place $\frac{n-1}{6}$ points each at $x_1 = (0, 0), x_2' = (0, 1 + 2\epsilon)$, $\frac{n-1}{12}$ points each at $x_1' = (0, \epsilon), x_2 = (0, 1 + \epsilon)$, $\frac{n-1}{4}$ points each at $y_1 = (1 + 0.9\epsilon, \epsilon), y_2 = (1 + \epsilon, 1 + 1.9\epsilon)$, and one point $z_1 = (0, 2)$ with $\epsilon = 0.3$. With some abuse of notation we will use the coordinate variables defined above to also denote the collection of points at their respective locations. We set $\mathcal{C}_3 = \{\{(x, y) \in S_2 \mid x \leq 0\}, \{(x, y) \in S_2 \mid x > 0\}\}$.

After merging the coincident points, all procedures will merge clusters $\tilde{x}_1 := \{x_1, x_1'\}$ and $\tilde{x}_2 := \{x_2, x_2', z_1\}$. Let's now consider the single linkage merge function. We have $D_{sgl}(\tilde{x}_1, \tilde{x}_2; d) = 1$ and all other cluster pairs are further apart. The next merge is therefore $\tilde{x} := \{\tilde{x}_1, \tilde{x}_2\}$. Also, $D_{sgl}(\tilde{x}, y_1; d) = 1 + 0.9\epsilon < \min\{D_{sgl}(\tilde{x}, y_2; d), D_{sgl}(y_1, y_2; d)\}$ leading to the merge $\{\tilde{x}, y_1\}$, and finally $y_2$ is merged in. A 2-pruning therefore has loss $\ell(T_{sgl}^{S_3}, \mathcal{C}_3) = \Omega(1)$. On the other hand, $D_{med}(\tilde{x}_1, \tilde{x}_2; d) = 1 + \epsilon > D_{med}(y_1, y_2; d) = \sqrt{(1 + 0.9\epsilon)^2 + 0.01\epsilon^2}$ and $D_{med}(\tilde{x}_1, y_1; d) = \sqrt{(1 + 0.9\epsilon)^2 + \epsilon^2} > \{D_{med}(y_1, y_2; d), D_{med}(\tilde{x}_1, \tilde{x}_2; d)\}$. As a result, median linkage would first

---

[11]A $k_0$-pruning for a tree $T$ is a partition of the points contained in $T$'s root into $k_0$ clusters such that each cluster is an internal node of $T$.

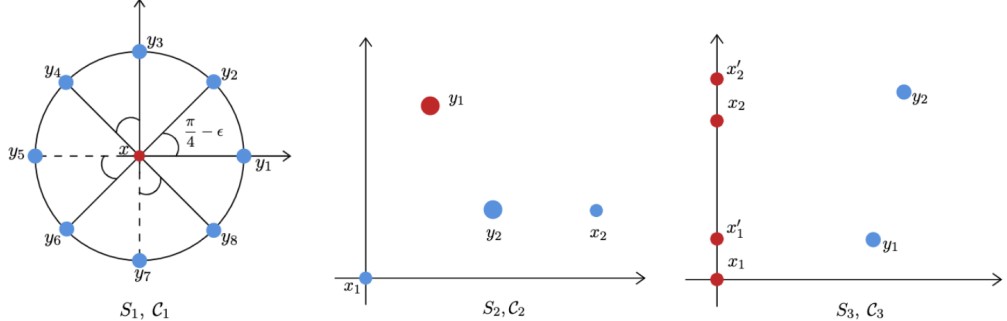

Figure 1: Construction of clustering instances showing the need for interpolating linkage heuristics. We give concrete instances and target clusterings where each of two-point based linkage procedures, i.e. single, complete and median linkage, dominates the other two.

merge $\{y_1, y_2\}$, followed by $\{\tilde{x}_1, \tilde{x}_2\}$, and 2-pruning yields $\mathcal{C}_3$. Complete linkage also merges $\{y_1, y_2\}$ first. But $D_{\text{cmpl}}(\tilde{x}_1, \tilde{x}_2; d) = 2 > D_{\text{cmpl}}(\tilde{x}_1, \{y_1, y_2\}; d)$. Thus, $\ell(T_{\text{cmpl}}^{S_3}, \mathcal{C}_3) = \Omega(1)$.

$\square$

# I  Additional details and proofs from Section 3

**Definition 4** (2-point-based merge function [BDL20])**. *A merge function $D$ is 2-point-based if for any pair of clusters $A, B \subseteq \mathcal{X}$ and any metric $d$, there exists a set of points $(a, b) \in A \times B$ such that $D(A, B; d) = d(a, b)$. Furthermore, the selection of $a$ and $b$ only depend on the relative ordering of the distances between points in $A$ and $B$. More formally, for any metrics $d$ and $d'$ such that $d(a, b) \leq d(a', b')$ if and only if $d'(a, b) \leq d'(a', b')$, then $D(A, B; d) = d(a, b)$ implies $D(A, B; d') = d'(a, b)$.*

For instance, single, median and complete linkage are 2-point-based, since the merge function $D(A, B; d)$ only depends on the distance $d(a, b)$ for some $a \in A, b \in B$. We have the following observation about our parameterized algorithm families $D_\rho^\Delta(A, B; \delta)$ when $\Delta$ consists of 2-point-based merge functions which essentially establishes piecewise structure with linear boundaries (in the sense of Definition 1).

**Lemma I.1.** *Suppose $S \in \Pi$ is a clustering instance, $\Delta$ is a set of 2-point-based merge functions with $|\Delta| = l$, and $\delta$ is a set of distance metrics with $|\delta| = m$. Consider the family of clustering algorithms with the parameterized merge function $D_\rho^\Delta(A, B; \delta)$. The corresponding dual class function $u_S(\cdot)$ is piecewise constant with $O(|S|^{4lm})$ linear boundaries.*

*Proof.* Let $(a_{ij}, b_{ij}, a'_{ij}, b'_{ij})_{1 \leq i \leq l, 1 \leq j \leq m} \subseteq S$ be sequences of $lm$ points each; for each such $a$, let $g_a : \mathcal{P} \to \mathbb{R}$ denote the function

$$g_a(\rho) = \sum_{i \in [l], d_j \in \delta} \alpha_{i,j}(\rho)(d_j(a_{ij}, b_{ij}) - d_j(a'_{ij}, b'_{ij}))$$

and let $\mathcal{G} = \{g_a \mid (a_{ij}, b_{ij}, a'_{ij}, b'_{ij})_{1 \leq i \leq l, 1 \leq j \leq m} \subseteq S\}$ be the collection of all such linear functions; notice that $|\mathcal{G}| = O(|S|^{4lm})$. Fix $\rho, \rho' \in \mathcal{P}$ with $g(\rho)$ and $g(\rho')$ having the same sign patterns for all such $g$. For each $A, B, A', B' \subseteq S$, $D_i \in \Delta$, and $d_j \in \delta$, we have $D_i(A, B; d_j) = d_j(a, b)$ and $D_i(A', B'; d_j) = d_j(a', b')$ for some $a, b, a', b' \in S$ (since $D_i$ is 2-point-based). Thus we can write $D_\rho(A, B; \delta) = \sum_{i \in [m], d_j \in \delta} \alpha_{i,j}(\rho) d_j(a_{ij}, b_{ij})$ for some $a_{ij}, b_{ij} \in S$; similarly, $D_\rho(A', B'; \delta) = \sum_{i \in [m], d_j \in \delta} \alpha_{i,j}(\rho) d_j(a'_{ij}, b'_{ij})$ for some $a'_{ij}, b'_{ij} \in S$. As a result, $D_\rho(A, B; \delta) \leq D_\rho(A', B'; \delta)$ if and only if

$$\sum_{i \in [l], d_j \in \delta} \alpha_{i,j}(\rho) \left(d_j(a_{ij}, b_{ij}) - d_j(a'_{ij}, b'_{ij})\right) \leq 0$$

which is exactly when $g_a(\rho) \leq 0$ for some sequence $a$. Since $g_a(\rho)$ and $g_a(\rho')$ have the same sign pattern, we have $D_\rho(A, B; \delta) \leq D_\rho(A', B'; \delta)$ if and only if $D_{\rho'}(A, B; \delta) \leq D_{\rho'}(A', B'; \delta)$. So $\rho$

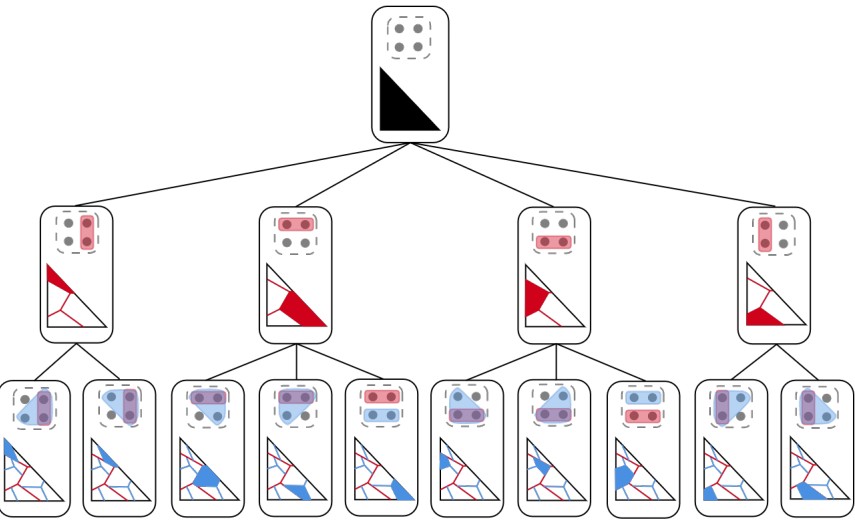

Figure 2: The first three levels of an example execution tree of a clustering instance on four points, with a two-parameter algorithm ($\mathcal{P} = \blacktriangle^2$). Successive partitions $\mathcal{P}_0, \mathcal{P}_1, \mathcal{P}_2$ are shown at merge levels 0, 1, and 2, respectively, and the nested shapes show cluster merges.

and $\rho'$ induce the same sequence of merges, meaning the algorithm's output is constant on each piece induced by $g$, as desired. $\qquad\square$

From Lemma I.1, we obtain a bound on the number of hyperplanes needed to divide $\mathcal{P}$ into output-constant pieces. Let $H$ be a set of hyperplanes which splits $\mathcal{P}$ into output-constant pieces; then, a naive approach to finding a dual-minimizing $\rho \in \mathcal{P}$ is to enumerate all pieces generated by $H$, requiring $O(|H|^d)$ runtime. However, by constructing regions merge-by-merge and successively refining the parameter space, we can obtain a better runtime bound which is output-sensitive in the total number of pieces.

*Proof of Lemma ??.* This is a simple corollary of Lemma I.1 for $m = 1$. In this case, we have $l = d + 1$. $\qquad\square$

**Lemma I.2.** *Consider the family of clustering algorithms with the parameterized merge function* $D^1_\rho(A, B; \delta)$. *Let $T^S_\rho$ denote the cluster tree computed using the parameterized merge function* $D^\Delta_\rho(A, B; d_0)$ *on sample $S$. Let $\mathcal{U}$ be the set of functions $\{u_\rho : S \mapsto \ell(T^S_\rho, \mathcal{C}) \mid \rho \in \mathbb{R}^d\}$ that map a clustering instance $S$ to $\mathbb{R}$. The dual class $\mathcal{U}^*$ is $(\mathcal{F}, |S|^4)$-piecewise decomposable, where* $\mathcal{F} = \{f_c : \mathcal{U} \to \mathbb{R} \mid c \in \mathbb{R}\}$ *consists of constant functions $f_c : u_\rho \mapsto c$.*

The key observation for the proof comes from [BDL20] where it is observed that two parameterized distance metrics $d_{\rho_1}, d_{\rho_2}$ behave identically (yield the same cluster tree) on a given dataset $S$ if the relative distance for all pairs of two points $(a, b), (a', b') \in S^2 \times S^2$, $d_\rho(a, b) - d_\rho(a', b')$, has the same sign for $\rho_1, \rho_2$. This corresponds to a partition of the parameter space with $|S|^4$ hyperplanes, with all distance metrics behaving identically in each piece of the partition. More formally, we have

*Proof of Lemma I.2.* Let $S$ be any clustering instance. Fix points $a, b, a', b' \in S$. Define the linear function $g_{a,b,a',b'}(\rho) = \sum_i \rho_i(d_i(a, b) - d_i(a', b'))$. If $d_\rho(\cdot, \cdot)$ denotes the interpolated distance metric, we have that $d_\rho(a, b) \leq d_\rho(a', b')$ if and only if $g_{a,b,a',b'}(\rho) \leq 0$. Therefore we have a set $H = \{g_{a,b,a',b'}(\rho) \leq 0 \mid a, b, a', b' \in S\}$ of $|S|^4$ hyperplanes such that in any piece of the sign-pattern partition of the parameter space by the hyperplanes, the interpolated distance metric behaves identically, i.e. for any $\rho, \rho'$ in the same piece $d_\rho(a, b) \leq d_\rho(a', b')$ iff $d_{\rho'}(a, b) \leq d_{\rho'}(a', b')$. The resulting clustering is therefore identical in these pieces. This means that for any connected

component R of $\mathbb{R}^d \setminus H$, there exists a real value $c_R$ such that $u_\rho(s_1, s_2) = c_R$ for all $\rho \in \mathbb{R}^d$. By definition of the dual, $u_{s_1,s_2}^*(u_\rho) = u_\rho(s_1, s_2) = c_R$. For each hyperplane $h \in H$, let $g^{(h)} \in \mathcal{G}$ denote the corresponding halfspace. Order these $k = |S|^4$ functions arbitrarily as $g_1, \ldots, g_k$. For a given connected component $R$ of $\mathbb{R}^d \setminus H$, let $\mathbf{b}_R \in \{0, 1\}^k$ be the corresponding sign pattern. Define the function $f^{(\mathbf{b}_R)} = f_{c_R}$ and for $\mathbf{b}$ not corresponding to any $R$, $f^{(\mathbf{b})} = f_0$. Thus, for each $\rho \in \mathbb{R}^d$,

$$u_{s_1,s_2}^*(u_\rho) = \sum_{\mathbf{b} \in \{0,1\}^k} \mathbb{I}\{g_i(u_\rho) = b_i \forall i \in [k]\} f^{(\mathbf{b})}(u_\rho).$$

$\square$

**Corollary I.3.** *For any clustering instance $S \in \Pi$, the dual class function $u_S(\cdot)$ for the family in Lemma I.2 is piecewise constant with $O\left(|S|^{4d}\right)$ pieces.*

**Lemma I.4.** *Let $S \in \Pi$ be a clustering instance, $\Delta$ be a set of merge functions, and $\delta$ be a set of distance metrics. Then, the corresponding dual class function $u_S(\cdot)$ is piecewise constant with $O(16^{|S|})$ linear boundaries of pieces.*

*Proof.* For each subset of points $A, B, A', B' \subseteq S$, let $g_{A,B,A',B'} : \mathcal{P} \to \mathbb{R}$ denote the function

$$g_{A,B,A',B'}(\rho) = D_\rho(A, B; \delta) - D_\rho(A', B'; \delta)$$

and let $\mathcal{G}$ be the collection of all such functions for distinct subsets $A, B, A', B'$. Observe that $\mathcal{G}$ is a class of linear functions with $|\mathcal{G}| \leq \left(2^{|S|}\right)^4 = 16^{|S|}$. Suppose that for $\rho, \rho' \in \mathcal{P}$, $g(\rho)$ and $g(\rho')$ have the same sign for all $g \in G$; then, the ordering over all cluster pairs $A, B$ of $D_\rho(A, B; \delta)$ is the same as that of $D_{\rho'}(A, B; \delta)$. At each stage of the algorithm, the cluster pair $A, B \subseteq S$ minimizing $D_\rho(A, B; \delta)$ is the same as that which minimizes $D_{\rho'}(A, B; \delta)$, so the sequences of merges produced by $\rho$ and $\rho'$ are the same. Thus the algorithm's output is constant on the region induced by $g_{A,B,A',B'}$, meaning $u_S(\cdot)$ is piecewise constant on the regions induced by $\mathcal{G}$, which have linear boundaries. $\square$

## I.1 Execution Tree

Formally we define an execution tree (Figure 2) as follows.

**Definition 5** (Execution tree)**.** *Let $S$ be a clustering instance with $|S| = n$, and $\emptyset \neq \mathcal{P} \subseteq [0, 1]^d$. The* execution tree *on $S$ with respect to $\mathcal{P}$ is a depth-$n$ rooted tree $T$, whose nodes are defined recursively as follows: $r = ([], \mathcal{P})$ is the root, where $[]$ denotes the empty sequence; then, for any node $v = ([u_1, u_2, \ldots, u_t], \mathcal{Q}) \in T$ with $t < n - 1$, the children of $v$ are defined as*

$$children(v) = \left\{ \left([u_1, u_2, \ldots, u_t, (A, B)], \mathcal{Q}_{A,B}\right) : \begin{array}{l} A, B \subseteq S \text{ is the } (t+1)^{st} \text{ merge by } \mathcal{A}_\rho \text{ for} \\ \text{exactly the } \rho \in \mathcal{Q}_{A,B} \subseteq \mathcal{P}, \text{ with } \emptyset \neq \mathcal{Q}_{A,B} \subseteq \mathcal{Q} \end{array} \right\}.$$

For an execution tree $T$ with $v \in T$ and each $i$ with $0 \leq i \leq n$, we let $\mathcal{P}_i$ denote the set of $\mathcal{Q}$ such that there exists a depth-$i$ node $v \in T$ and a sequence of merges $\mathcal{M}$ with $v = (\mathcal{M}, \mathcal{Q})$. Intuitively, the execution tree represents all possible execution paths (i.e. sequences for the merges) for the algorithm family when run on the instance $S$ as we vary the algorithm parameter $\rho \in \mathcal{P}$. Furthermore, each $\mathcal{P}_i$ is a subdivision of the parameter space into pieces where each piece has the first $i$ merges constant. We establish the execution tree captures all possible sequences of merges by some algorithm $\mathcal{A}_\rho$ in the parameterized family via its nodes, and each node corresponds to a convex polytope if the parameter space $\mathcal{P}$ is a convex polytope (Lemmata I.5 and I.6).

Our cell enumeration algorithm for computing all the pieces of the dual class function now simply computes the execution tree, using Algorithm 1 to compute the children nodes for any given node, starting with the root.

**Lemma I.5.** *Let $S$ be a clustering instance and $T$ be its execution tree with respect to $\mathcal{P}$. Then, if a sequence of merges $\mathcal{M} = [u_1, u_2, \ldots, u_t]$ is attained by $\mathcal{A}_\rho$ for some $\rho \in \mathcal{P}$, then there exists some $v \in T$ at depth $t$ with $v = (\mathcal{M}, \mathcal{Q})$ and with $\mathcal{Q} \subseteq \mathcal{P}$ being the exact set of values of $\rho$ for which $\mathcal{A}_\rho$ may attain $\mathcal{M}$. Conversely, for every node $v = (\mathcal{M}, \mathcal{Q}) \in T$, $\mathcal{M}$ is a valid sequence of merges attainable by $\mathcal{A}_\rho$ for some $\rho \in \mathcal{P}$.*

*Proof.* We proceed by induction on $t$. For $t = 0$, the only possible sequence of merges is the empty sequence, which is obtained for all $\rho \in \mathcal{P}$. Furthermore, the only node in $T$ at depth 0 is the root $([], \mathcal{P})$, and the set $\mathcal{P}$ is exactly where an empty sequence of merges occurs.

Now, suppose the claim holds for some $t \geq 0$. We show both directions in the induction step.

For the forward direction, let $\mathcal{M}_{t+1} = [u_1, u_2, \ldots, u_t, u_{t+1}]$, and suppose $\mathcal{M}_{t+1}$ is attained by $\mathcal{A}_\rho$ for some $\rho \in \mathcal{P}$. This means that $\mathcal{M}_t = [u_1, u_2, \ldots, u_t]$ is attained by $\mathcal{A}_\rho$ as well; by the induction hypothesis, there exists some node $v_t = (\mathcal{M}_t, \mathcal{Q}_t) \in T$ at depth $t$, where $\rho \in \mathcal{Q}_t$ and $\mathcal{Q}_t$ is exactly the set of values for which $\mathcal{A}$ may attain $\mathcal{M}_t$. Now, $u_{t+1}$ is a possible next merge by $\mathcal{A}_\rho$ for some $\rho \in \mathcal{Q}_t$; by definition of the execution tree, this means $v_t$ has some child $v_{t+1} = (\mathcal{M}_{t+1}, \mathcal{Q}_{t+1})$ in $T$ such that $\mathcal{Q}_{t+1}$ is the set of values where $u_{t+1}$ is the next merge in $\mathcal{Q}_t$. Moreover, $\mathcal{Q}_{t+1}$ is exactly the set of values $\rho \in \mathcal{P}$ for which $A_\rho$ can attain the merge sequence $\mathcal{M}_{t+1}$. In other words for any $\rho' \in \mathcal{P} \setminus \mathcal{Q}_{t+1}$, $A_{\rho'}$ cannot attain the merge sequence $\mathcal{M}_{t+1}$. Otherwise, either some $\rho' \in \mathcal{P} \setminus \mathcal{Q}_t$ attains $\mathcal{M}_{t+1}$, meaning $A_{\rho'}$ attains $\mathcal{M}_t$ (contradicting the induction hypothesis), or $A_{\rho'}$ attains $\mathcal{M}_{t+1}$ for some $\rho' \in \mathcal{Q}_{t+1} \setminus \mathcal{Q}_t$, contradicting the definition of $\mathcal{Q}_{t+1}$.

For the backward direction, let $v_{t+1} = (\mathcal{M}_{t+1}, \mathcal{Q}_{t+1}) \in T$ at depth $t + 1$. Since $v_{t+1}$ is not the root, $v_{t+1}$ must be the child of some node $v_t$, which has depth $t$. By the induction hypothesis, $v_t = (\mathcal{M}_t, \mathcal{Q}_t)$, where $\mathcal{M}_t = [u_1, u_2, \ldots, u_t]$ is attained by $\mathcal{A}_\rho$ for some $\rho \in \mathcal{P}$. Thus by definition of the execution tree, $\mathcal{M}_{t+1}$ has the form $[u_1, u_2, \ldots, u_t, (A, B)]$, for some merging of cluster pairs $(A, B)$ which is realizable for $\rho \in \mathcal{Q}_t$. Thus $\mathcal{M}_{t+1}$ is a valid sequence of merges attainable by $\mathcal{A}_\rho$ for some $\rho \in \mathcal{P}$. □

**Lemma I.6.** *Let $S$ be a clustering instance and $T$ be its execution tree with respect to $\mathcal{P}$. Suppose $\mathcal{P}$ is a convex polytope; then, for each $v = (\mathcal{M}, \mathcal{Q}) \in T$, $\mathcal{Q}$ is a convex polytope.*

*Proof.* We proceed by induction on the tree depth $t$. For $t = 0$, the only node is $([], \mathcal{P})$, and $\mathcal{P}$ is a convex polytope. Now, consider a node $v \in T$ at depth $t + 1$; by definition of the execution tree, $v = (\mathcal{M}_v, \mathcal{Q}_v)$ is the child of some node $u \in T$, where the depth of $u$ is $t$. Inductively, we know that $w = (\mathcal{M}_w, \mathcal{Q}_w)$, for some convex polytope $\mathcal{Q}_w$. We also know $\mathcal{M}_w$ has the form $\mathcal{M}_w = [u_1, u_2, \ldots, u_t]$, and thus $\mathcal{Q}_v$ is defined to be the set of points $\rho \in \mathcal{Q}_w$ where the merge sequence $\mathcal{M}_v = [u_1, u_2, \ldots, u_t, (A, B)]$ is attainable for some fixed $A, B \subseteq S$. Notice that the definition of being attainable by the algorithm $\mathcal{A}_\rho$ is that $D_\rho(A, B; \delta)$ is minimized over all choices of next cluster pairs $A', B'$ to merge. That is, $\mathcal{Q}_v$ is the set of points

$$\mathcal{Q}_v = \{\rho \in \mathcal{Q}_w \mid D_\rho(A, B; \delta) \leq D_\rho(A', B'; \delta) \text{ for all available cluster pairs } A', B' \text{ after } \mathcal{M}_w\}$$

Since $D_\rho(A, B; \delta)$ is an affine function of $\rho$, the constraint $D_\rho(A, B; \delta) \leq D_\rho(A', B'; \delta)$ is a half-space. In other words, $\mathcal{Q}_v$ is the intersection of a convex polytope $\mathcal{Q}_w$ with finitely many half space constraints, meaning $\mathcal{Q}_v$ is itself a convex polytope. □

It follows from Lemma I.6 that $\mathcal{P}_i$ forms a convex subdivision of $\mathcal{P}$, where each $\mathcal{P}_{i+1}$ is a refinement of $\mathcal{P}_i$; Figure 2 (in the appendix) shows an example execution tree corresponding to a partition of a 2-dimensional parameter space. From Lemma I.5, the sequence of the first $i$ merges stays constant on each region $P \in \mathcal{P}_i$. Our algorithm computes a representation of the execution tree of an instance $S$ with respect to $\mathcal{P}$; to do so, it suffices to provide a procedure to list the children of a node in the execution tree. Then, a simple breadth-first search from the root will enumerate all the leaves in the execution tree.

Now, our goal is to subdivide $P$ into regions in which the $(j+1)^{\text{st}}$ merge is constant. Each region corresponds to a cluster pair being merged at step $j + 1$. Since we know these regions are always convex polytopes (Lemma I.6), we can provide an efficient algorithm for enumerating these regions.

Our algorithm provides an output-sensitive guarantee by ignoring the cluster pairs which are never merged. Supposing there are $n_t$ unmerged clusters, we start with some point $x \in P$ and determine which piece $W$ it is in. Then, we search for more non-empty pieces contained in $P$ by listing the "neighbors" of $W$. The neighbors of $W$ are pieces inside $P$ that are adjacent to $W$; to this end, we will more formally define a graph $G_P$ associated with $P$ where each vertex is a piece and two vertices have an edge when the pieces are adjacent in space. Then we show that we can enumerate neighbors of a vertex efficiently and establish that $G_P$ is connected. It follows that listing the pieces is simply a matter of running a graph search algorithm from one vertex of $G_P$, thus only incurring a cost for each *non-empty piece* rather than enumerating through all $n_t^4$ pairs of pieces.

*Proof of Corollary 3.2.* The key observation is that on any iteration $i$, the number of adjacencies $H_i = O(R_i)$. This is because for any region $P \in \mathcal{P}_i$, $P$ is a polygon divided into convex subpolygons, and the graph $G_P$ has vertices which are faces and edges which cross between faces. Since the subdivision of $P$ can be embedded in the plane, so can the graph $G_P$. Thus $G_P$ is planar, meaning $H_i = O(R_i)$. Plugging into Theorem 3.1, noting that $(n - i + 1)^2 \leq n^2$, $H_i \leq H$, and $R_i \leq R$, we obtain the desired runtime bound of $O\left(\sum_{i=1}^n (R + RT_M)n^2\right) = O(RT_M n^3)$. $\qquad\square$

## I.2   Auxiliary lemmas and proofs of runtime bounds for our algorithm

**Lemma I.7.** *Fix an affine function $f : \mathbb{R} \to \mathbb{R}^d$ via $f(x) = xa + b$, for $a, b \in \mathbb{R}^d$ and $a \neq 0^d$. For a subset $S \subseteq \mathbb{R}$, if $f(S)$ is convex and closed, then $S$ is also convex and closed.*

*Proof.* First note that $f$ is injective, since $a \neq 0^d$. To show convexity of $S$, take arbitrary $x, y \in S$ and $\lambda \in [0, 1]$; we show that $\lambda x + (1 - \lambda)y \in S$. Consider $f(\lambda x + (1 - \lambda)y)$:

$$f(\lambda x + (1 - \lambda)y) = (\lambda x + (1 - \lambda)y)a + b$$
$$= \lambda(xa + b) + (1 - \lambda)(ya + b)$$

By definition, $ya + b, xa + b \in f(S)$, so it follows that $f(\lambda x + (1 - \lambda)y) \in f(S)$ by convexity of $f(S)$. So there exists some $z \in S$ with $f(z) = f(\lambda x + (1 - \lambda)y)$, but since $f$ is injective, $\lambda x + (1 - \lambda)y = z \in S$. Thus $S$ is convex.

To show closedness of $S$, we show $\mathbb{R} \setminus S$ is open. Let $N(x, r)$ denote the open ball of radius $r$ around $x$, in either one-dimensional or $d$-dimensional space. Let $x \in \mathbb{R} \setminus S$; we know $f(x) \notin f(S)$ since $f$ is injective. Since $\mathbb{R}^d \setminus f(S)$ is open, there exists some $r > 0$ with $N(f(x), r) \subseteq \mathbb{R}^d \setminus f(S)$. Then, take $\mathbf{e} = \frac{r}{\|a\|_2} > 0$; for every $y \in N(x, \mathbf{e})$, we have

$$\|f(x) - f(y)\|_2 = \|xa + b - ya - b\|_2 < |x - y|\|a\|_2 \leq r$$

and so $f(y) \in N(f(x), r) \subseteq \mathbb{R}^d \setminus f(S)$, meaning $y \notin S$ since $f$ is injective. Thus $N(x, \mathbf{e}) \subseteq R \setminus S$, meaning $S$ is closed as desired. $\qquad\square$

This allows us to prove the following key lemma. We describe a proof sketch first. For arbitrary $(i, j), (i', j') \in V_P^*$, we show that there exists a path from $(i, j)$ to $(i', j')$ in $G_P$. We pick arbitrary points $w \in Q_{i,j}, x \in Q_{i',j'}$; we can do this because by definition, $V_P^*$ only has elements corresponding to non-empty cluster pairs. Then, we draw a straight line segment in $P$ from $w$ to $x$. When we do so, we may pass through other sets on the way; each time we pass into a new region, we traverse an edge in $G_P$, so the sequence of regions we pass through on this line determines a $G_P$-path from $w$ to $x$.

**Lemma I.8.** *The vertices $V_P^*$ of the region adjacency graph $G_P$ form a connected component; all other connected components of $G_P$ are isolated vertices.*

*Proof.* It suffices to show that for arbitrary vertices $(i_1, j_1), (i_2, j_2) \in V_P$, there exists a path from $(i_1, j_1)$ to $(i_2, j_2)$ in $G_P$. For ease of notation, define $Q_1 = Q_{i_1,j_1}$ and $Q_2 = Q_{i_2,j_2}$.

Fix arbitrary points $u \in Q_1$ and $w \in Q_2$. If $u = w$ then we're done, since the edge from $(i_1, j_1)$ to $(i_2, j_2)$ exists, so suppose $u \neq w$. Consider the line segment $L$ defined as

$$L = \{\lambda u + (1 - \lambda)w : \lambda \in [0, 1]\}$$

Since $Q_1, Q_2 \subseteq P$, we have $u, w \in P$. Furthermore, by convexity of $P$, it follows that $L \subseteq P$.

Define the sets $R_{i,j}$ as

$$R_{i,j} = Q_{i,j} \cap L$$

Since each $Q_{i,j}$ and $L$ are convex and closed, so is each $R_{i,j}$. Furthermore, since $\bigcup_{i,j} Q_{i,j} = P$, we must have $\bigcup_{i,j} R_{i,j} = L$. Finally, define the sets $S_{i,j}$ as

$$S_{i,j} = \{t \in [0, 1] : tu + (1 - t)w \in R_{i,j}\} \subseteq [0, 1]$$

Note that $S_{i,j}$ is convex and closed; the affine map $f : S_{i,j} \to R_{i,j}$ given by $f(x) = xu + (1-x)w = x(u-w) + w$ has $R_{i,j}$ as an image. Furthermore, $u - w \neq 0^d$; by Lemma I.7, the preimage $S_{i,j}$ must be convex and closed. Furthermore, $\bigcup_{i,j} S_{i,j} = [0,1]$.

The only convex, closed subsets of $[0,1]$ are closed intervals. We sort the intervals in increasing order based on their lower endpoint, giving us intervals $I_1, I_2, \ldots, I_\ell$. We also assume all intervals are non-empty (we throw out empty intervals). Let $\sigma(p)$ denote the corresponding cluster pair associated with interval $I_p$; that is, if the interval $I_p$ is formed from the set $S_{i,j}$, then $\sigma(p) = (i,j)$.

Define $a_i, b_i$ to be the lower and upper endpoints, respectively, of $I_i$. We want to show that for all $1 \leq i \leq \ell - 1$, the edge $\{\sigma(i), \sigma(i+1)\} \in E_P$; this would show that $\sigma(1)$ is connected to $\sigma(\ell)$ in the $V_P$. But $\sigma(1) = (i_1, j_1)$ and $\sigma(\ell) = (i_2, j_2)$, so this suffices for our claim.

Now consider intervals $I_i = [a_i, b_i]$ and $I_{i+1} = [a_{i+1}, b_{i+1}]$. It must be the case that $b_i = a_{i+1}$; otherwise, some smaller interval would fit in the range $[b_i, a_{i+1}]$, and it would be placed before $I_{i+1}$ in the interval ordering.

Since $b_i \in I_i \cap I_{i+1}$, by definition, $ub_i + (1-b_i)w \in R_{\sigma(i)} \cap R_{\sigma(i+1)}$. In particular, $ub_i + (1-b_i)w \in Q_{\sigma(i)} \cap Q_{\sigma(i+1)}$; by definition of $E_P$, this means $\{\sigma(i), \sigma(i+1)\} \in E_P$, as desired. $\qquad\square$

**Theorem 3.1.** *Let $S$ be a clustering instance with $|S| = n$, and let $R_i = |\mathcal{P}_i|$ and $R = R_n$. Let $H_t = \left|\{(\mathcal{Q}_1, \mathcal{Q}_2) \in \mathcal{P}_t^2 \mid \mathcal{Q}_1 \cap \mathcal{Q}_2 \neq \emptyset\}\right|$ denote the total number of adjacencies between any two pieces of $\mathcal{P}_i$ and $H = H_n$. Then, the leaves of the execution tree on $S$ can be computed in time $\tilde{O}\left(\sum_{i=1}^n (H_i + R_i T_M)(n - i + 1)^2\right)$, where $T_M$ is the time to compute the merge function.*

*Proof.* Let $T$ be the execution tree with respect to $S$, and let $T_t$ denote the vertices of $T$ at depth $t$. From Theorem 2.2, for each node $v = (\mathcal{M}, \mathcal{Q}) \in T$ with depth $t$, we can compute the children of $v$ in time $O(n_t^2 \cdot \text{LP}(d, E_v) + V_v \cdot n_t^2 K)$, where $V_v$ is the number of children of $v$, and

$$
E_v = \left|\left\{ \begin{array}{c} (\mathcal{Q}_1, \mathcal{Q}_2) \in \mathcal{P}_{t+1}^2 \mid \mathcal{Q}_1 \cap \mathcal{Q}_2 \neq \emptyset \text{ and} \\ u_1 = (\mathcal{M}_1, \mathcal{Q}_1), u_2 = (\mathcal{M}_2, \mathcal{Q}_2) \\ \text{for some children } u_1, u_2 \text{ of } v \end{array} \right\}\right|.
$$

Now, observe

$$
\sum_{v \in T_{t+1}} E_v \leq H_{t+1}
$$

since $H_{t+1}$ counts all adjacent pieces $\mathcal{Q}_1, \mathcal{Q}_2$ in $\mathcal{P}_{t+1}$; each pair is counted at most once by some $E_v$. Similarly, we have $\sum_{v \in T_{t+1}} V_v \leq R_{t+1}$, since $R_{t+1}$ counts the total size of $\mathcal{P}_{t+1}$. Note that $n_{t+1} = (n-t)$, since $t$ merges have been executed by time $t+1$, so $n_i = n-i+1$. Seidel's algorithm is a randomized algorithm that may be used for efficiently solving linear programs in low dimensions, the expected running time for solving an LP in $d$ variables and $m$ constraints is $O(d! \cdot E_v)$ (also holds with high probability, e.g. Corollary 2.1 of [Sei91]). There are also deterministic algorithms with the same (in fact slightly better) worst-case runtime bounds [Cha18]. Therefore, we can set $\text{LP}(d, E_v) = O(d! \cdot E_v)$. So that the total cost of computing $\mathcal{P}_i$ is

$$
O\left(\sum_{i=1}^n \sum_{v \in T_i} d! \cdot E_v (n - i + 1)^2 + V_v K (n - i + 1)^2\right) = O\left(\sum_{i=1}^n (d! \cdot H_i + R_i K)(n - i + 1)^2\right)
$$

as desired. $\qquad\square$

# J   Further details and proofs from Section 4

We provide further details for Algorithm 5 in Appendix J.2, and other proofs from Section 4 are located in Appendix J.3.

## J.1   Example dynamic programs for sequence alignment

We exhibit how two well-known sequence alignment formulations can be solved using dynamic programs which fit our model in Section 4. In Section J.1.1 we show a DP with two free parameters ($d = 2$), and in Section J.1.2 we show another DP which has three free parameters ($d = 3$).

### J.1.1 Mismatches and spaces

Suppose we only have two features, *mismatches* and *spaces*. The alignment that minimizes the cost $c$ may be obtained using a dynamic program in $O(mn)$ time. The dynamic program is given by the following recurrence relation for the cost function which holds for any $i, j > 0$, and for any $\rho = (\rho_1, \rho_2)$,

$$
C(s_1[:i], s_2[:j], \rho) = \begin{cases} C(s_1[:i-1], s_2[:j-1], \rho) & \text{if } s_1[i] = s_2[j], \\[2mm] \begin{aligned} \min \big\{ & \rho_1 + C(s_1[:i-1], s_2[:j-1], \rho), \\ & \rho_2 + C(s_1[:i-1], s_2[:j], \rho), \\ & \rho_2 + C(s_1[:i], s_2[:j-1], \rho) \quad \big\} \end{aligned} & \text{if } s_1[i] \neq s_2[j]. \end{cases}
$$

The base cases are $C(\phi, \phi, \rho) = 0, C(\phi, s_2[:j], \rho) = j\rho_2, = C(s_1[:i], \phi, \rho) = i\rho_2$ for $i, j \in [m] \times [n]$. Here $\phi$ denotes the empty sequence. One can write down a similar recurrence for computing the optimal alignment $\tau(s_1, s_2, \rho)$.

We can solve the non base-case subproblems $(s_1[:i], s_2[:j])$ in any non-decreasing order of $i+j$. Note that the number of cases $V = 2$, and the maximum number of subproblems needed to compute a single DP update $L = 3$ ($L_1 = 1, L_2 = 3$). For a non base-case problem (i.e. $i, j > 0$) the cases are given by $q(s_1[:i], s_2[:j]) = 1$ if $s_1[i] = s_2[j]$, and $q(s_1[:i], s_2[:j]) = 2$ otherwise. The DP update in each case is a minimum of terms of the form $c_{v,l}(\rho, (s_1[:i], s_2[:j])) = \rho \cdot w_{v,l} + \sigma_{v,l}(\rho, (s_1[:i], s_2[:j]))$. For example if $q(s_1[:i], s_2[:j]) = 2$, we have $w_{2,1} = \langle 1, 0 \rangle$ and $\sigma_{2,1}(\rho, (s_1[:i], s_2[:j]))$ equals $C(s_1[:i-1], s_2[:j-1], \rho)$, i.e. the solution of previously solved subproblem $(s_1[:i-1], s_2[:j-1])$, the index of this subproblem depends on $l, v$ and index of $(s_1[:i], s_2[:j])$ but not on $\rho$ itself.

### J.1.2 Mismatches, spaces and gaps

Suppose we have three features, *mismatches*, *spaces* and *gaps*. Typically gaps (consecutive spaces) are penalized in addition to spaces in this model, i.e. the cost of a sequence of three consecutive gaps in an alignment $(\ldots a - - - b \ldots, \ldots a' \ p \ q \ r \ b' \ldots)$ would be $3\rho_2 + \rho_3$ where $\rho_2, \rho_3$ are costs for *spaces* and *gaps* respectively [KKW10]. The alignment that minimizes the cost $c$ may again be obtained using a dynamic program in $O(mn)$ time. We will need a slight extension of our DP model from Section 4 to capture this. We have three subproblems corresponding to any problem in $\Pi_{s_1, s_2}$ (as opposed to exactly one subproblem, which was sufficient for the example in J.1.1). We have a set of subproblems $\pi(s_1, s_2)$ with $|\pi(s_1, s_2)| \leq 3|\Pi_{s_1, s_2}|$ for which our model is applicable. For each $(s_1[:i], s_2[:j])$ we can compute the three costs (for any fixed $\rho$)

- $C_s(s_1[:i], s_2[:j], \rho)$ is the cost of optimal alignment that ends with substitution of $s_1[i]$ with $s_2[j]$.
- $C_i(s_1[:i], s_2[:j], \rho)$ is the cost of optimal alignment that ends with insertion of $s_2[j]$.
- $C_d(s_1[:i], s_2[:j], \rho)$ is the cost of optimal alignment that ends with deletion of $s_1[i]$.

The cost of the overall optimal alignment is simply $C(s_1[:i], s_2[:j], \rho) = \min\{C_s(s_1[:i], s_2[:j], \rho), C_i(s_1[:i], s_2[:j], \rho), C_d(s_1[:i], s_2[:j], \rho)\}$.

The dynamic program is given by the following recurrence relation for the cost function which holds for any $i, j > 0$, and for any $\rho = (\rho_1, \rho_2, \rho_3)$,

$$
C_s(s_1[:i], s_2[:j], \rho) = \min \begin{cases} \rho_1 + C_s(s_1[:i-1], s_2[:j-1], \rho), \\ \rho_1 + C_i(s_1[:i-1], s_2[:j-1], \rho), \\ \rho_1 + C_d(s_1[:i-1], s_2[:j-1], \rho) \end{cases}
$$

$$
C_i(s_1[:i], s_2[:j], \rho) = \min \begin{cases} \rho_2 + \rho_3 + C_s(s_1[:i], s_2[:j-1], \rho), \\ \rho_2 + C_i(s_1[:i], s_2[:j-1], \rho), \\ \rho_2 + \rho_3 + C_d(s_1[:i], s_2[:j-1], \rho) \end{cases}
$$

$$
C_d(s_1[:i], s_2[:j], \rho) = \min \begin{cases} \rho_2 + \rho_3 + C_s(s_1[:i-1], s_2[:j], \rho), \\ \rho_2 + \rho_3 + C_i(s_1[:i-1], s_2[:j], \rho), \\ \rho_2 + C_d(s_1[:i-1], s_2[:j], \rho) \end{cases}
$$

---

**Algorithm 5:** COMPUTECOMPACTEXECUTIONDAG

---

 1: **Input**: Execution DAG $G_e = (V_e, E_e)$, problem instance $(s_1, s_2)$
    $\mathcal{P}_0 \leftarrow \mathcal{P}$
 2: $v_1, \ldots, v_n \leftarrow$ topological ordering of vertices $V_e$
 3: **for** $i = 1$ to $n$ **do**
 4:     Let $S_i$ be the set of nodes with incoming edges to $v_i$
 5:     For $v_s \in S_i$, let $\mathcal{P}_s$ denote the partition corresponding to $v_s$
 6:     $\mathcal{P}_i \leftarrow$ COMPUTEOVERLAYDP$(\{\mathcal{P}_s \mid s \in S_i\})$
 7:     **for** each $p \in \mathcal{P}_i$ **do**
 8:        $p' \leftarrow$ COMPUTESUBDIVISIONDP$(p, (s_1, s_2))$
 9:        $\mathcal{P}_i \leftarrow \mathcal{P}_i \setminus \{p\} \cup p'$
10:     $\mathcal{P}_i \leftarrow$ RESOLVEDEGENERACIESDP$(\mathcal{P}_i)$
11: **return** Partition $\mathcal{P}_n$

---

By having three subproblems for each $(s_1[:i], s_2[:j])$ and ordering the non base-case problems again in non-decreasing order of $i + j$, the DP updates again fit our model (1).

## J.2 Details of the Execution-DAG based algorithm

We start with some well-known terminology from computational geometry.

**Definition 6.** *A (convex) subdivision $S$ of $P \subseteq \mathbb{R}^d$ is a finite set of disjoint $d$-dimensional (convex) sets (called cells) whose union is $P$. The overlay $S$ of subdivisions $S_1, \ldots, S_n$ is defined as all nonempty sets of the form $\bigcap_{i \in [n]} s_i$ with $s_i \in S_i$. With slight abuse of terminology, we will refer to closures of cells also as cells.*

The COMPUTEOVERLAY procedure takes a set of partitions, which are convex polytopic subdivisions of $\mathbb{R}^d$, and computes their overlay. We will represent a convex polytopic subdivision as a list of cells, each represented as a list of bounding hyperplanes. Now to compute the overlay of subdivisions $P_1, \ldots, P_L$, with lists of cells $\mathcal{C}_1, \ldots, \mathcal{C}_L$ respectively, we define $|\mathcal{C}_1| \times \cdots \times |\mathcal{C}_L|$ sets of hyperplanes $H_{j_1, \ldots, j_L} = \{\bigcup_{l \in [L]} \mathcal{H}(c_{j_l}^{(l)})\}$, where $c_{j_l}^{(l)}$ is the $j_l$-th cell of $P_l$ and $\mathcal{H}(c)$ denotes the hyperplanes bounding cell $c$. We compute the cells of the overlay by applying Clarkson's algorithm [Cla94] to each $H_{j_1, \ldots, j_L}$. We have the following guarantee about the running time of Algorithm 6.

---

**Algorithm 6:** COMPUTEOVERLAYDP

---

**Input**: Convex polytopic subdivisions $P_1, \ldots, P_L$ of $\mathbb{R}^d$, represented as lists $\mathcal{C}_j$ of hyperplanes
 for each cell in the subdivision
$\mathcal{H}(c_{j_l}^{(l)}) \leftarrow$ hyperplanes bounding $j_l$-th cell of $P_l$ for $l \in [L], j_l \in \mathcal{C}_l$
**for** *each* $j_1, \ldots, j_L \in |\mathcal{C}_1|, \ldots, |\mathcal{C}_L|$ **do**
    $H_{j_1, \ldots, j_L} \leftarrow \{\bigcup_{l \in [L]} \mathcal{H}(c_{j_l}^{(l)})\}$
    $H'_{j_1, \ldots, j_L} \leftarrow$ CLARKSON$(H_{j_1, \ldots, j_L})$
$\mathcal{C} \leftarrow$ non-empty lists of hyperplanes in $H'_{j_1, \ldots, j_L}$ for $j_l \in \mathcal{C}_l$
**return** Partition represented by $\mathcal{C}$

---

**Lemma J.1.** *Let $R_{i,j}$ denote the number of pieces in $P[i][j]$, and $\tilde{R} = \max_{i \leq m, j \leq n} P[i][j]$. There is an implementation of the* COMPUTEOVERLAYDP *routine in Algorithm 5 which computes the overlay of $L$ convex polytopic subdivisions in time $O(L\tilde{R}^{L+1} \cdot \mathrm{LP}(d, \tilde{R}^L + 1))$, which is $O(d!L\tilde{R}^{2L+1})$ using algorithms for solving low-dimensional LPs [Cha18].*

*Proof.* Consider Algorithm 6. We apply the Clarkson's algorithm at most $\tilde{R}^L$ times, once corresponding to each $L$-tuple of cells from the $L$ subdivisions. Each iteration corresponding to cell $c$ in the output overlay $\mathcal{O}$ (corresponding to $\mathcal{C}$) has a set of at most $L\tilde{R}$ hyperplanes and yields at most $R_c$ non-redundant hyperplanes. By Theorem 2.1, each iteration takes time $O(L\tilde{R} \cdot \mathrm{LP}(d, R_c + 1))$, where $\mathrm{LP}(d, R_c + 1)$ is bounded by $O(d!R_c)$ for the algorithm of [Cha18]. Note that $\sum_c R_c$ corresponds to

---

**Algorithm 7:** COMPUTESUBDIVISIONDP

---

**Input**: Convex Polytope $P$, problem instance $(s_1, s_2)$
$v \leftarrow$ the DP *case* $q((s_1, s_2))$ for the problem instance
$\rho_0 \leftarrow$ an arbitrary point in $P$
$(t_1^0, t_2^0) \leftarrow$ optimal alignment of $(s_1, s_2)$ for parameter $\rho_0$, using subproblem
$\quad (s_1[:i_{v,l_0}], s_2[:j_{v,l_0}])$ for some $l_0 \in [L_v]$
$\texttt{mark} \leftarrow \emptyset$, $\texttt{polys} \leftarrow$ new hashtable, $\texttt{poly\_queue} \leftarrow$ new queue
$\texttt{poly\_queue.enqueue}(l_0)$
**while** $poly\_queue.non\_empty()$ **do**
    $l \leftarrow \texttt{poly\_queue.dequeue}()$;
    Continue to next iteration if $l \in \texttt{mark}$
    $\texttt{mark} \leftarrow \texttt{mark} \cup \{l\}$
    $\mathcal{L} \leftarrow P$
    **for** *all subproblems* $(s_1[:i_{v,l_1}], s_2[:j_{v,l_1}])$ *for* $l_1 \in [L_v], l_1 \neq l$ **do**
        Add the half-space inequality $b^T \rho \leq c$ corresponding to
        $c_{v,l}(\rho, (s_1, s_2)) \leq c_{v,l_1}(\rho, (s_1, s_2))$ to $\mathcal{L}$ ; /* Label the constraint $(b, c)$ with
        $l_1$ */
    $I \leftarrow \text{CLARKSON}(\mathcal{L})$
    $\texttt{poly\_queue.enqueue}(l')$
    for each $l'$ such that the constraint labeled by it is in $I$
    $\texttt{polys}[l] \leftarrow \{\mathcal{L}[\ell] : \ell \in I\}$
**return** $\texttt{polys}$

---

the total number of edges in the cell adjacency graph of $\mathcal{O}$, which is bounded by $\tilde{R}^{2L}$. Further note that $R_c \leq \tilde{R}^L$ for each $c \in \mathcal{C}$ and $|\mathcal{C}| \leq \tilde{R}^L$ to get a runtime bound of $O(L\tilde{R}^{L+1} \cdot \text{LP}(d, \tilde{R}^L + 1))$. $\qquad \square$

We now consider an implementation for the COMPUTESUBDIVISIONDP subroutine. The algorithm computes the hyperplanes across which the subproblem used for computing the optimal alignment changes in the recurrence relation (1) by adapting Algorithm 1. We restate the algorithm in the context of sequence alignment as Algorithm 7.

**Lemma J.2.** *Let $R_{i,j}$ denote the number of pieces in $P[i][j]$, and $\tilde{R} = \max_{i \leq m, j \leq n} R_{i,j}$. There is an implementation of* COMPUTESUBDIVISIONDP *routine in Algorithm 5 with running time at most $O((L^{2d+2} + L^{2d}\tilde{R}^L) \cdot \text{LP}(d, L^2 + \tilde{R}^L))$ for each outer loop of Algorithm 5. If the algorithm of [Cha18] is used to solve the LP, this is at most $O(d! L^{2d+2} \tilde{R}^{2L})$.*

*Proof.* Consider Algorithm 7. For any piece $p$ in the overlay, all the required subproblems have a fixed optimal alignment, and we can find the subdivision of the piece by adapting Algorithm 1 (using $O(L^2 + \tilde{R}^L)$ hyperplanes corresponding to subproblems and piece boundaries). The number of pieces in the subdivision is at most $L^{2d}$ since we have at most $L^2$ hyperplanes intersecting the piece, so we need $O(L^{2d+2} + L^{2d}\tilde{R}^L)$ time to list all the pieces $\mathcal{C}_p$. The time needed to run Clarkson's algorithm is upper bounded by $O(\sum_{c \in \mathcal{C}_p} (L^2 + \tilde{R}^L) \cdot \text{LP}(d, R_c + 1)) = O(\sum_{c \in \mathcal{C}_p} (L^2 + \tilde{R}^L) \cdot \text{LP}(d, L^2 + \tilde{R}^L)) = O((L^{2d+2} + L^{2d}\tilde{R}^L) \cdot \text{LP}(d, L^2 + \tilde{R}^L))$. Using [Cha18] to solve the LP, this is at most $O(d! \tilde{R}^{2L} L^{2d+4})$. $\qquad \square$

**Lemma J.3.** *Let $R_{i,j}$ denote the number of pieces in $P[i][j]$, and $\tilde{R} = \max_{i \leq m, j \leq n} R_{i,j}$. There is an implementation of* RESOLVEDEGENERACIESDP *routine in Algorithm 5 with running time at most $O(\tilde{R}^{2L} L^{4d})$ for each outer loop of Algorithm 5.*

*Proof.* The RESOLVEDEGENERACIESDP is computed by a simple BFS over the cell adjacency graph $G_c = (V_c, E_c)$ (i.e. the graph with polytopic cells as nodes and edges between polytopes sharing facets). We need to find (maximal) components of a subgraph of the cell adjacency graph where each node in the same component has the same optimal alignment. This is achieved by a simple BFS in $O(|V_c| + |E_c|)$ time. Indeed, by labeling each polytope with the corresponding optimal alignment, we can compute the components of the subgraph of $G_c$ with edges restricted

to nodes joining the same optimal alignment. Note that the resulting polytopic subdivision after the merge is still a convex subdivision using arguments in Lemma J.5, but applied to appropriate sequence alignment subproblem. As noted in the proof of Lemma J.2, we have $|V_c| \leq L^{2d}\tilde{R}^L$ since the number of cells within each piece $p$ is at most $L^{2d}$ and there are at most $\tilde{R}^L$ pieces in the overlay. Since $|E_c| \leq |V_c|^2$, we have an implementation of RESOLVEDEGENERACIESDP in time $O((L^{2d}\tilde{R}^L)^2) = O(\tilde{R}^{2L}L^{4d})$. □

Finally we can put all the above together to give a proof of Theorem 4.1.

*Proof of Theorem 4.1.* The proof follows by combining Lemma J.1, Lemma J.2 and Lemma J.3. Note that in the execution DAG, we have $|V_e| \leq |E_e| = O(T_{\text{DP}})$. Further, we invoke COMPUTEOVERLAYDP and RESOLVEDEGENERACIESDP $|V_e|$ times across all iterations and COMPUTESUBDIVISIONDP across the $|V_e|$ outer loops. □

## J.3 Additional Proofs

The following results closely follow and extend the corresponding results from [BDD$^+$21]. Specifically, we generalize to the case of two sequences of unequal length, and provide sharper bounds on the number of distinct alignments and boundary functions in the piecewise decomposition (even in the special case of equal lengths). We first have a bound on the total number of distinct alignments.

**Lemma J.4.** *For a fixed pair of sequences $s_1, s_2 \in \Sigma^m \times \Sigma^n$, with $m \leq n$, there are at most $m(m + n)^m$ distinct alignments.*

*Proof.* For any alignment $(t_1, t_2)$, by definition, we have $|t_1| = |t_2|$ and for all $i \in [|t_1|]$, if $t_1[i] = -$, then $t_2[i] \neq -$ and vice versa. This implies that $t_1$ has exactly $n - m$ more gaps than $t_2$. To prove the upper bound, we count the number of alignments $(t_1, t_2)$ where $t_2$ has exactly $i$ gaps for $i \in [m]$. There are $\binom{n+i}{i}$ choices for placing the gap in $t_2$. Given a fixed $t_2$ with $i$ gaps, there are $\binom{n}{n-m+i}$ choices for placing the gap in $t_1$. Thus, there are at most $\binom{n+i}{i}\binom{n}{n-m+i} = \frac{(n+i)!}{i!(m-i)!(n-m+i)!} \leq (m + n)^m$ possibilities since $i \leq m$. Summing over all $i$, we have at most $m(m + n)^m$ alignments of $s_1, s_2$. □

This implies that the dual class functions are piecewise-structured in the following sense.

**Lemma J.5.** *Let $\mathcal{U}$ be the set of functions $\{u_\rho : (s_1, s_2) \mapsto u(s_1, s_2, \rho) \mid \rho \in \mathbb{R}^d\}$ that map sequence pairs $s_1, s_2 \in \Sigma^m \times \Sigma^n$ to $\mathbb{R}$ by computing the optimal alignment cost $C(s_1, s_2, \rho)$ for a set of features $(l_i(\cdot))_{i\in[d]}$. The dual class $\mathcal{U}^*$ is $(\mathcal{F}, \mathcal{G}, m^2(m + n)^{2m})$-piecewise decomposable, where $\mathcal{F} = \{f_c : \mathcal{U} \to \mathbb{R} \mid c \in \mathbb{R}\}$ consists of constant functions $f_c : u_\rho \mapsto c$ and $\mathcal{G} = \{g_w : \mathcal{U} \to \{0, 1\} \mid w \in \mathbb{R}^d\}$ consists of halfspace indicator functions $g_w : u_\rho \mapsto \mathbb{I}\{w \cdot \rho < 0\}$.*

*Proof.* Fix a pair of sequences $s_1$ and $s_2$. Let $\tau$ be the set of optimal alignments as we range over all parameter vectors $\rho \in \mathbb{R}^d$. By Lemma J.4, we have $|\tau| \leq m(m + n)^m$. For any alignment $(t_1, t_2) \in \tau$, the algorithm $A_\rho$ will return $(t_1, t_2)$ if and only if

$$\sum_{i=1}^{d} \rho_i l_i(s_1, s_2, t_1, t_2) > \sum_{i=1}^{d} \rho_i l_i(s_1, s_2, t_1', t_2')$$

for all $(t_1', t_2') \in \tau \setminus \{(t_1, t_2)\}$. Therefore, there is a set $H$ of at most $\binom{|\tau|}{2} \leq m^2(m + n)^{2m}$ hyperplanes such that across all parameter vectors $\rho$ in a single connected component of $\mathbb{R}^d \setminus H$, the output of the algorithm $A_\rho$ on $(s_1, s_2)$ is fixed. This means that for any connected component R of $\mathbb{R}^d \setminus H$, there exists a real value $c_R$ such that $u_\rho(s_1, s_2) = c_R$ for all $\rho \in \mathbb{R}^d$. By definition of the dual, $u^*_{s_1,s_2}(u_\rho) = u_\rho(s_1, s_2) = c_R$. For each hyperplane $h \in H$, let $g^{(h)} \in \mathcal{G}$ denote the corresponding halfspace. Order these $k = \binom{|\tau|}{2}$ functions arbitrarily as $g_1, \ldots, g_k$. For a given connected component $R$ of $\mathbb{R}^d \setminus H$, let $\mathbf{b}_R \in \{0, 1\}^k$ be the corresponding sign pattern. Define the function $f^{(\mathbf{b}_R)} = f_{c_R}$ and for $\mathbf{b}$ not corresponding to any $R$, $f^{(\mathbf{b})} = f_0$. Thus, for each $\rho \in \mathbb{R}^d$,

$$u^*_{s_1,s_2}(u_\rho) = \sum_{\mathbf{b}\in\{0,1\}^k} \mathbb{I}\{g_i(u_\rho) = b_i \forall i \in [k]\}f^{(\mathbf{b})}(u_\rho).$$

$\square$

For the special case of $d = 2$, we have an algorithm which runs in time $O(RT_{\mathrm{DP}})$, where $R$ is the number of pieces in $P[m][n]$ which improves on the prior result $O(R^2 + RT_{\mathrm{DP}})$ for two-parameter sequence alignment problems. The algorithm employs the ray search technique of [Meg78] (also employed by [GBN94] but for more general sequence alignment problems) and enjoys the following runtime guarantee.

**Theorem J.6.** *For the global sequence alignment problem with $d = 2$, for any problem instance $(s_1, s_2)$, there is an algorithm to compute the pieces for the dual class function in $O(RT_{\mathrm{DP}})$ time, where $T_{\mathrm{DP}}$ is the time complexity of computing the optimal alignment for a fixed parameter $\rho \in \mathbb{R}^2$, and $R$ is the number of pieces of $u_{(s_1, s_2)}(\cdot)$.*

*Proof.* We note that for any alignment $(t_1, t_2)$, the boundary functions for the piece where $(t_1, t_2)$ is an optimal alignment are straight lines through the origin of the form

$$\rho_1 l_1(s_1, s_2, t_1, t_2) + \rho_2 l_2(s_1, s_2, t_1, t_2) > \rho_1 l_1(s_1, s_2, t_1', t_2') + \rho_2 l_2(s_1, s_2, t_1', t_2')$$

for some alignment $(t_1', t_2')$ different from $(t_1, t_2)$. The intersection of these halfplanes is either the empty set or the region between two straight lines through the origin. The output subdivision therefore only consists of the axes and straight lines through the origin in the positive orthant.

We will present an algorithm using the ray search technique of [Meg78]. The algorithm computes the optimal alignment $(t_1, t_2)$, $(t_1', t_2')$ at points $\rho = (0, 1)$ and $\rho = (1, 0)$. If the alignments are identical, we conclude that $(t_1, t_2)$ is the optimal alignment everywhere. Otherwise, we find the optimal alignment $(t_1'', t_2'')$ for the intersection of line $L$ joining $\rho = (0, 1)$ and $\rho = (1, 0)$, with the line $L'$ given by

$$\rho_1 l_1(s_1, s_2, t_1, t_2) + \rho_2 l_2(s_1, s_2, t_1, t_2) > \rho_1 l_1(s_1, s_2, t_1', t_2') + \rho_2 l_2(s_1, s_2, t_1', t_2')$$

If $(t_1'', t_2'') = (t_1, t_2)$ or $(t_1'', t_2'') = (t_1', t_2')$, we have exactly 2 optimal alignments and the piece boundaries are given by $L'$ and the axes. Otherwise we repeat the above process for alignment pairs $(t_1'', t_2''), (t_1', t_2')$ and $(t_1'', t_2''), (t_1, t_2)$. Notice we need to compute at most $R + 1$ dynamic programs to compute all the pieces, giving the desired time bound. $\square$

