# OpenReview forum: "Accelerating ERM for data-driven algorithm design using output-sensitive techniques"
_NeurIPS.cc/2024/Conference — NeurIPS 2024 poster_

### Official Review · Reviewer_YVeC · 2024-06-17

**Soundness:** 2
**Presentation:** 2
**Contribution:** 3
**Rating:** 5
**Confidence:** 2

**Summary:**

This paper addresses the problem of learning optimal parameters for data-driven algorithm design. A characteristic of the problem is that the dual loss function, which measures the performance of an algorithm as a function of parameters, is discontinuous. Nevertheless, the dual loss is typically piecewise structured (constant, linear, etc.) with linear boundaries. Thus, roughly speaking, the problem of finding optimal parameters reduces to exploring polytopic cells partitioned by boundary hyperplanes.

The main contribution is a cell enumeration algorithm that runs in output-polynomial time. The algorithm can be seen as a breadth-first search on a cell adjacency graph, where the enumeration of neighbors is done in an output-sensitive manner based on Clarkson's algorithm. The resulting output-sensitive complexity can be significantly better than the worst-case, as demonstrated in Example 1.

The authors then instantiate the ERM method based on the cell enumeration for linkage-based clustering and DP-based sequence alignment. The applications involve designing execution graphs, which originate from the execution tree of [BDL20]. Combining appropriate problem-specific execution graphs with the cell enumeration leads to the improved time complexity of ERM in several data-driven algorithm design problems, as in Table 1.

**Strengths:**

1. The paper addresses the important problem of optimizing algorithm parameters in data-driven algorithm design.
2. The theoretical results given in Table 1 appear strong compared with previous ones.
3. The output-sensitive cell enumeration might be of independent interest in the context of computational geometry.

**Weaknesses:**

1. The paper would have been more appealing if implementations of the proposed methods and experimental results were provided.
2. The paper is somewhat dense and it is not easy to follow the technical details.

**Questions:**

1. I would like to know more intuition about how AugmentedClarkson differs from the original Clarkson and why it is important.
2. While the paper focuses on linear boundaries, some studies on data-driven algorithm design consider parameter spaces partitioned by polynomials:

https://proceedings.neurips.cc/paper_files/paper/2022/hash/db2cbf43a349bc866111e791b58c7bf4-Abstract-Conference.html

https://proceedings.mlr.press/v178/bartlett22a.html

https://proceedings.mlr.press/v206/sakaue23a.html

Is there a possibility of applying similar enumeration ideas to such situations?

**Limitations:**

Yes.

---

> ### Author Rebuttal · Authors · 2024-08-07
>
> We thank the reviewer for their time and insightful comments. Re experiments, we note that prior empirical work already suggests usefulness of output-sensitive guarantees on typical instances, which we elaborate below.
>
> ``Experiments:``
> Our work is motivated from prior empirical research (lines 91-104) which indicate that the output cell size on typical problems is empirically much smaller than the worst-case cell size. The missing component is the development of algorithms with theoretical runtime guarantees which are output-sensitive, as previously proposed algorithms scale with the worst-case cell size. We expect implementation of our (LP-based) algorithm to be relatively straightforward, and would like to remark that there are no reasonable baselines known as previous algorithms were intractable.
>
> ``Beyond linear boundaries:``
> We agree that algebraic boundaries would be the next direction to look at based on our work. One simple way to apply our algorithms to the polynomial boundary case is to use linearization by projecting into a higher dimension corresponding to the monomials (this would work if the polynomials have a small constant degree e.g. Thm 3.1, Lem 4.1 in [1] or Lemma 2 in [2]). A more direct extension which tries to compute cells induced by polynomial boundaries in an output-sensitive way is an interesting direction for future work.
>
> ``AugmentedClarkson:``
> Clarkson’s Algorithm computes the set of non-redundant hyperplanes in a linear system in an output-sensitive time complexity. Intuitively, our augmentation additionally keeps track of the neighboring cells corresponding to the non-redundant hyperplanes to facilitate the breadth-first search over the cell adjacency graph. This is important as directly applying Clarkson’s and searching for the nearest cell (in a fixed direction) across each bounding hyperplane can add to the runtime complexity.
>
> *References*
>
> [1] Balcan, Maria-Florina, Siddharth Prasad, Tuomas Sandholm, and Ellen Vitercik. "Structural analysis of branch-and-cut and the learnability of gomory mixed integer cuts." Advances in Neural Information Processing Systems 35 (2022): 33890-33903.
>
> [2] Balcan, Maria-Florina, Travis Dick, and Manuel Lang. "Learning to Link." In International Conference on Learning Representations 2020.

---

> > ### Comment · Reviewer_YVeC · 2024-08-12
> >
> > Thank you for the detailed response. I appreciate the answers to my questions. I retain my score.

---

### Official Review · Reviewer_swPv · 2024-07-05

**Soundness:** 3
**Presentation:** 2
**Contribution:** 2
**Rating:** 6
**Confidence:** 4

**Summary:**

In data-driven algorithm design, we are given a collection of problem instances sampled from an unknown distribution, and a family of algorithms for solving the problem, typically parameterized by a real-valued multivariate parameter. The goal is to find a setting of parameters such that the performance of the algorithm they parameterize is close to optimal among the family of algorithms, in expectation over the distribution of instances. Most prior work is firstly focused on the generalization aspect, i.e., on showing that a small number of samples suffices (in the sample complexity sense) for finding approximately optimal parameters for the distribution (these are the ERM parameters for the given sample of instance), and thus the family of algorithms is "learnable". It then (sometimes) proceeds to develop an efficient algorithm for finding those ERM parameters, based on the structure used to prove the generalization bound.

This paper focuses more systematically on the ERM efficiency aspect. To this end, it starts with a common theme in many prior works on data-driven algorithms, that had been abstracted out and formulated in a generalized form in Balcan et al. (STOC 2021): for any fixed problem instance, the function that maps a setting of parameters to utility (of invoking their associated algorithm on that instance) admits a simple piecewise structure. Say, there is a small number of "simple" boundary functions (say, linear thresholds) that induce a partition of the parameter space R^d such that the utility function restricted to each piece is "simple" (say, constant). This is helpful in bounding the VC dimension of the utility functions and thus proving generalization bounds, and also potentially for navigating the parameter space efficiently to find the ERM parameters.

The novelty in this paper is an attempt to give a more systematic recipe for the second part (navigating the piecewise structure for efficient ERM), with two main advantages -- (1) creating a unified framework that takes care of some parts of the ERM procedure in a general way, thus restricting the portion that needs to be figured out per each problem individually, and (2) obtaining algorithms whose running time depends on the actual number of pieces in the given instances ("output-sensitive") rather than worst-case number of pieces. The per-problem part to figure out is a subroutine that, given a problem instance and parameter setting p, returns a list of candidate boundary functions for the piece that contains p, and this subroutine depends on the specific problem in question. The unified part of the framework uses this subroutine to search through the pieces in an "output-sensitive" running time.

__Post-rebuttal__: I appreciate the authors' elaborations on the technical content, and the conceptual aspect of the paper. I have raised my score to support acceptance.

**Strengths:**

The strength of this paper is that the matter of efficient ERM in data-driven algorithms indeed merits its own systematic study rather than being left as an afterthought of the generalization bounds.

**Weaknesses:**

The main weakness is that the end result isn't very strong: the framework is restricted to linear boundary functions and (more disconcertingly) to a constant number of parameters, and for the most part does not yield improved running times in the worst case, but a different notion of efficiency (output sensitivity). It tends more to systematically organizing ideas that have appeared in the literature in one form or another and less to introducing new algorithmic insights or techniques. I also feel that the presentation and writing could be too opaque for a wide readership like that of NeurIPS.

**Questions:**

N/A

---

> ### Author Rebuttal · Authors · 2024-08-07
>
> We respectfully disagree with the reviewer that the main result is not strong. Since its inception (around 2016), the field of theoretical guarantees of data driven algorithm design has been focused on sample complexity results. The *major* direction that has been left open in this field is to also consider the computational efficiency of data-driven algorithms. Our paper is the first to consider this question at some level of generality. We note that one cannot hope for the same level of generality as for the sample complexity of data driven algorithm design since algorithmic consideration have always been much more problem specific even in the classic (non-data driven) scenarios, that is in classic learning theory: even in classic learning theory we have general sample complexity results for ERM, yet the computational complexity of ERM algorithms depend heavily on the class (and even for basic classes we still have major open questions).
>
> Our work contains many subtle aspects, for example:
> - We make a novel connection with computational geometry to give beyond worst-case improvements on runtime efficiency. It is a priori not clear  which tool to use to advance the state-of-the-art here.
> - The execution tree approach for linkage clustering needs several technical lemmas for establishing soundness of the approach (e.g. lemmas I.5 to I.8 in Appendix I).
> - The execution DAG approach used in sequence alignment is a completely novel contribution of this paper, and is critical to achieve the desired output-sensitive runtime guarantee.
>
> ``Applications of our framework (linear boundaries, constant parameters):`` We show several distinct applications in our work where the linear boundary and constant parameter size is relevant for the data-driven algorithm design problem: linkage clustering - both metric learning and distance function learning; sequence alignment; two-part tariff pricing (App F); item-pricing with anonymous prices (App F). The linear boundary setting appears more frequently than one might initially expect, see e.g. [1] for several applications. Moreover, by a simple linearization argument, our approach yields ERM implementations for polynomial boundaries with small degrees e.g.  e.g. Thm 3.1, Lem 4.1 in [2] or Lemma 2 in [3]. Finally, we remark that we initiate a detailed study of computationally efficient implementation of ERM for data-driven algorithm design and tackle the natural first case where we already show significant improvement over previous runtimes (Table 1) provided output size is small.
>
> ``the most part does not yield improved running times in the worst case, but a different notion of efficiency (output sensitivity)``
> Output-sensitivity is a relevant notion for data-driven algorithm design. We summarize motivation from prior empirical work in lines 91-104, which indicates usefulness of output-sensitivity from a practical perspective. Note that the whole point of data-driven algorithm design is to provide beyond worst-case improvements in algorithm design [4], so it is not unusual to expect a beyond worst-case measure of efficiency (output-sensitivity and fixed parameter tractability).
>
> ``It tends more to systematically organizing ideas that have appeared in the literature in one form or another and less to introducing new algorithmic insights or techniques``
> We remark that using a computational geometry based technique in the context of learning theory is in itself remarkable and has hardly ever appeared in prior literature. Moreover, a direct application of Clarkson's algorithm itself is not always sufficient. Our “execution DAG” technique proposed for the sequence alignment problem is a novel algorithmic idea not from any previous literature, and is crucial for tracking the partition of the parameter space for each DP sub-problem. In this case the number of alternative alignments (and therefore the number of relevant hyperplanes $t_{LRS}$) for any fixed optimal alignment is exponential, so a direct application of Clarkson’s algorithm would still be exponential runtime.
>
> *References*
>
> [1] Balcan, Maria-Florina, Tuomas Sandholm, and Ellen Vitercik. "A general theory of sample complexity for multi-item profit maximization." In Proceedings of the 2018 ACM Conference on Economics and Computation, pp. 173-174. 2018.
>
> [2] Balcan, Maria-Florina, Siddharth Prasad, Tuomas Sandholm, and Ellen Vitercik. "Structural analysis of branch-and-cut and the learnability of gomory mixed integer cuts." Advances in Neural Information Processing Systems 35 (2022): 33890-33903.
>
> [3] Balcan, Maria-Florina, Travis Dick, and Manuel Lang. "Learning to Link." In International Conference on Learning Representations 2020.
>
> [4] Maria-Florina Balcan. Data-Driven Algorithm Design. In Tim Roughgarden, editor, Beyond Worst Case Analysis of Algorithms. Cambridge University Press, 2020.

---

> > ### Comment · Reviewer_swPv · 2024-08-11
> >
> > Thank you for your response. I have read it and it will be considered.

---

### Official Review · Reviewer_gq34 · 2024-07-12

**Soundness:** 3
**Presentation:** 3
**Contribution:** 3
**Rating:** 7
**Confidence:** 2

**Summary:**

The paper explores computational aspects of implementing ERM in data-driven algorithm design.

The paper contributes an efficient algorithm to ennumerate cells induced by a collection of hyperplanes. The paper then shows how to utilize this as a subprocedure to solve ERM problems for algorithm design, focusing on linkage-based clustering, sequence alignment, and two-part tariff pricing.

**Strengths:**

One of the main interesting things I find about this paper is that the runtime of the ERM implementations is instance dependent, and specifically depends on the number of sum dual class loss function pieces. The paper comments that their runtime bounds imply improvements over prior work in the worst-case R but also can be faster for "typical" R.

The paper is well-written and easy to follow. The paper discusses relevant background and related work.

**Weaknesses:**

To what extent is the approach generalizable to other data-driven algorithm design problems? Is there a generic principle or a general characterization of the problems for which this approach can be utilized?

**Questions:**

See questions above.

**Limitations:**

Yes.

---

> ### Author Rebuttal · Authors · 2024-08-07
>
> We thank the reviewer for their time and useful comments.
>
> ``Generalizability to other data-driven algorithm design problems:`` Our approach is applicable to a fairly large number of problems, for example the various mechanism design problems in [1] are (d,t)-delineable which is a special case of Definition 1 (We include a couple examples in Appendix F). Our current approach is useful whenever the loss as a function of the algorithmic parameter on a fixed problem instance can be shown to have a piecewise structure with linear transition boundaries. The execution tree/DAG technique enables output-sensitive runtime complexity when the piecewise structure can be viewed as a refinement of pieces induced in successive algorithmic steps (e.g. see Figure 2), each refinement involving linear decision boundaries in the parameter space, and we expect this to be useful beyond the linkage clustering and sequence alignment problems considered. There is also potential for extending our work beyond linear transition boundaries e.g. to algebraic boundaries which would be relevant for several problems of interest in data-driven algorithm design.
>
>
>
> [1] Balcan, Maria-Florina, Tuomas Sandholm, and Ellen Vitercik. "A general theory of sample complexity for multi-item profit maximization." In Proceedings of the 2018 ACM Conference on Economics and Computation, pp. 173-174. 2018.

---

> > ### Comment · Reviewer_gq34 · 2024-08-09
> > **Response to Rebuttal**
> >
> > Thank you for addressing my question. I keep my score.

---

### Author Rebuttal · Authors · 2024-08-07

Our work is a first major step in making the growing field of data-driven algorithm design a practically feasible endeavor by addressing the frequently noted open question of computational complexity [1, 2]. Our proposed algorithms provide concrete and significant improvements in the running time of ERM from $n^{O(d)}$ (formally XP time) to $f(d).poly(n)$ effectively establish fixed parameter tractability (for parameter $d$, input size $n$, see Table 1) assuming output is polynomially large. Our work is well-motivated by prior empirical work on clustering and sequence alignment (lines 91-104) which show that typical output size is dramatically smaller than worst-case bounds, implying that our methods lead to dramatic speed-up over prior work on typical instances.

We have shown four different applications of our results (linear transition boundaries, constant number of parameters) in three different domains (data science, computational biology, computational economics), which is rare for algorithmic results that use the specific structure of the problem. We further expect more direct and indirect applications of our results [3, 4]. Technically, our work involves several novel ideas:
- Use of computational geometry in learning theory does not have precedent in the literature as far as we are aware. We identify and adapt the relevant techniques to address the crucial problem of implementing ERM in data-driven algorithm design.
- We develop novel techniques (execution tree and DAG) which are critical in addition to comp. geo. tools, e.g. directly using Clarkson’s algorithm would still lead to exponential time in $n$ but our techniques circumvent this.
- We present our results in a modular framework, reducing the ERM problem to a simpler problem that can be solved in a problem-specific way, making it easier to apply our results for future applications.


*References*

[1] Rishi Gupta and Tim Roughgarden. Data-driven algorithm design. Communications of the ACM, 63(6):87–94, 2020.

[2] Avrim Blum, Chen Dan, and Saeed Seddighin. Learning complexity of simulated annealing. In International Conference on Artificial Intelligence and Statistics (AISTATS), pages 1540–1548. PMLR, 2021.

[3] Balcan, Maria-Florina, Tuomas Sandholm, and Ellen Vitercik. "A general theory of sample complexity for multi-item profit maximization." In Proceedings of the 2018 ACM Conference on Economics and Computation, pp. 173-174. 2018.

[4] Balcan, Maria-Florina, Siddharth Prasad, Tuomas Sandholm, and Ellen Vitercik. "Structural analysis of branch-and-cut and the learnability of gomory mixed integer cuts." Advances in Neural Information Processing Systems 35 (2022): 33890-33903.

---

### Decision · Program_Chairs · 2024-09-25

**Decision:**

Accept (poster)

**Comment:**

During the discussion, the reviewers unanimously acknowledged the conceptual contribution of the paper in pioneering a systematic study of the computational complexity of ERMs. The rebuttal helped clarify some initial concerns, further strengthening the case for acceptance. I recommend acceptance.